EMBO
Molecular Medicine

# mGluR7 allosteric modulator AMN082 corrects protein synthesis and pathological phenotypes in FXS

Vipendra Kumar [1], Kwan Young Lee [1], Anirudh Acharya[1], Matthew S Babik[1], Catherine A Christian-Hinman [1,2,3], Justin S Rhodes[2,3,4] & Nien-Pei Tsai [1,2,3]✉

## Abstract

**Fragile X syndrome (FXS) is the leading cause of inherited autism and intellectual disabilities. Aberrant protein synthesis due to the loss of fragile X messenger ribonucleoprotein (FMRP) is the major defect in FXS, leading to a plethora of cellular and behavioral abnormalities. However, no treatments are available to date. In this study, we found that activation of metabotropic glutamate receptor 7 (mGluR7) using a positive allosteric modulator named AMN082 represses protein synthesis through ERK1/2 and eIF4E signaling in an FMRP-independent manner. We further demonstrated that treatment of AMN082 leads to a reduction in neuronal excitability, which in turn ameliorates audiogenic seizure susceptibility in *Fmr1* KO mice, the FXS mouse model. When evaluating the animals' behavior, we showed that treatment of AMN082 reduces repetitive behavior and improves learning and memory in *Fmr1* KO mice. This study uncovers novel functions of mGluR7 and AMN082 and suggests the activation of mGluR7 as a potential therapeutic approach for treating FXS.**

**Keywords** FXS; FMRP; mGluR7; Autism; Protein Synthesis
**Subject Categories** Genetics, Gene Therapy & Genetic Disease; Neuroscience

## Introduction

Fragile X syndrome (FXS) is monogenic and inherited and is the most prevalent form of autism. It affects 1:4000 males and 1:8000 females (Hagerman et al, 2009; Budimirovic and Kaufmann, 2011; Turner et al, 1996). Major clinical symptoms of FXS involve intellectual disability, hyperarousal, hyperactivity, and seizures (Hagerman et al, 2017; Hagerman and Hagerman, 2022; Berry-Kravis et al, 2010). FXS is caused by the lack of fragile X messenger ribonucleoprotein (FMRP) that is encoded by the *Fmr1* gene. In FXS, the *Fmr1* gene is transcriptionally silenced due to abnormally expanded CGG repeats (>200) that lead to DNA hypermethylation.

FMRP is primarily involved in repression of protein synthesis by directly or indirectly interfering with translating mRNAs (Hagerman and Hagerman, 2022). A growing body of evidence arising from various model systems of FXS has confirmed that exaggerated protein synthesis is central to most disease-specific molecular and behavioral abnormalities in FXS (Bolduc et al, 2008; Till et al, 2015; Bhakar et al, 2012; Osterweil et al, 2010; Raj et al, 2021). Many strategies aimed at rebalancing basally elevated protein synthesis in FXS have been introduced (Dölen and Bear, 2008; Michalon et al, 2012; Sharma et al, 2010; Gurney et al, 2017; McCamphill et al, 2020). Among them, inhibition of metabotropic glutamate receptor 5 (mGluR5) is most extensively studied as multiple mGluR5 antagonists have been shown to exhibit disease-modifying potential by causing a reduction in protein synthesis and correction in behavioral abnormalities in FXS animal models, such as *Fmr1* KO mice. However, the unfortunate failure of clinical trials, such as the one for the negative allosteric modulator (NAM) against mGluR5, Mavoglurant (Scharf et al, 2015; Berry-Kravis et al, 2016), has prompted the search for alternative therapeutic strategies.

mGluR7, an understudied mGluR, belongs to group III mGluRs along with other members, namely mGluR4, mGluR6, and mGluR8. Although group III mGluRs have been shown to participate in synaptic plasticity via ERK/MAPK signaling (Dasgupta et al, 2020), the detailed function and mechanism of individual group III mGluRs remain unclear. Subcellular localization studies have shown that mGluR7 can be located in the presynaptic active zones of glutamatergic and GABAergic neurons in the somatosensory cortex and hippocampus (Shigemoto et al, 1997; Dalezios et al, 2002) as well as postsynaptic membranes of glutamatergic neurons in the prefrontal cortex (Gu et al, 2012). mGluR7 is encoded by *Grm7* gene, a known autism-linked gene whose truncation and missense mutations have been associated with idiopathic autism and developmental delay (Yang and Pan, 2013; Fisher et al, 2018). Studies have demonstrated that mGluR7 can function by causing a decrease in excitatory neurotransmission through inhibition of glutamate release from the presynaptic terminal (Palazzo et al, 2016). Based on this logic, other studies have also shown that mice deficient in mGluR7 (*mGluR7* knockout [KO]) exhibit increased seizure susceptibility (Sansig et al, 2001). Additionally, *mGluR7* KO mice also show deficits in neuronal plasticity and working memory (Hölscher et al, 2005). Despite these

[1]Department of Molecular and Integrative Physiology, School of Molecular and Cellular Biology, University of Illinois at Urbana-Champaign, Urbana, IL 61801, USA. [2]Neuroscience Program, University of Illinois at Urbana-Champaign, Urbana, IL 61801, USA. [3]Beckman Institute for Advanced Science and Technology, University of Illinois at Urbana-Champaign, Urbana, IL 61801, USA. [4]Department of Psychology, University of Illinois at Urbana-Champaign, Champaign, IL 61820, USA. ✉E-mail: nptsai@illinois.edu

prior studies, the molecular mechanism by which mGluR7 achieves its physiological functions and whether activation of mGluR7 can be a potential therapeutic approach for neurodevelopmental disorders, such as FXS, remains unclear.

In this study, we showed that activation of mGluR7 using a positive allosteric modulator, N,N′-dibenzhydrylethane-1,2-diamine, AMN082, causes a reduction in protein synthesis via extracellular signal-regulated kinase 1/2- and eukaryotic translation initiation factor 4E (ERK1/2 and eIF4E, respectively)-associated signaling in an *Fmr1*-independent manner. Furthermore, we found that activation of mGluR7 leads to a significant reduction in neuronal excitability and audiogenic seizure (AGS) phenotype in *Fmr1* KO mice. Additionally, mGluR7 activation by AMN082 leads to a significant reduction in repetitive behavior and improvement in learning and memory in *Fmr1* KO mice. Together, these findings reveal a novel mechanism underlying the physiological effects of mGluR7 through translational control and suggest activation of mGluR7 as a potential therapeutic approach for treating FXS.

## Results

### Activation of mGluR7 reduces protein synthesis in both WT and *Fmr1* KO neurons

mGluR7 is highly expressed in multiple brain regions, including hippocampus, neocortex, and hypothalamus (Kinzie et al, 1995; Bradley et al, 1996; Ohishi et al, 1995). To begin, we first aimed to characterize the expression levels and patterns of mGluR7 between WT and *Fmr1* KO mice. We performed immunohistochemical staining of mGluR7 using an antibody against mGluR7a on brain sections from post-natal (P) day-60 male WT and *Fmr1* KO mice. No commercial antibodies suitable for immunohistochemical staining are available against mGluR7b, the other major isoform of mGluR7. As shown in Fig. 1A, we did not observe any visible change in mGluR7a in the sub-regions of the hippocampus, including CA3, CA1, and dentate gyrus between WT and *Fmr1* KO mice. Antibody specificity was confirmed using brain sections from *mGluR7* KO mice. We next measured the expression of mGluR7a and mGluR7b in the forebrain lysates of WT and *Fmr1* KO mice by western blotting. As shown in Fig. 1B, consistent with our observation using brain sections, we did not find any significant changes in the total levels of mGluR7a or mGluR7b between WT and *Fmr1* KO mice. Interestingly, when performing surface protein biotinylation in WT and *Fmr1* KO primary cortical neuron cultures followed by western blotting, we observed a slight but significant decrease in surface mGluR7a and an increase in surface mGluR7b in *Fmr1* KO neurons when compared with WT neurons (Fig. 1C). It should be noted that cortical neurons were used to obtain enough cells for surface biotinylation. Together, these data suggest a slight alteration of surface expression of mGluR7 isoforms in *Fmr1* KO neurons.

To begin exploring the effect of mGluR7 on protein synthesis, we treated WT and *Fmr1* KO cortical neurons with a selective mGluR7 allosteric agonist, AMN082 (1 μM) or antagonist 6-(4-Methoxyphenyl)-5-methyl-3-(4-pyridinyl)-isoxazolo[4,5-c]pyridin-4(5H)-one (MMPIP, 1 μM) for 2 h and employed the surface sensing of translation (SUnSET) technique to label newly synthesized protein with puromycin (10 μg/ml) during the last

hour of treatment. Puromycin-labeled proteins were detected by western blotting using an anti-puromycin antibody. As shown in Fig. 1D, we observed a significant increase in the protein synthesis in *Fmr1* KO neurons compared to WT neurons, as has been observed previously (Dölen et al, 2007). Treatment of AMN082 was able to cause significant reduction in protein synthesis in *Fmr1* KO cultures to a level similar to basal WT levels. Because AMN082 treatment also significantly reduces protein synthesis in WT cultures, it suggests that AMN082 acts through *Fmr1*-independent mechanism in this novel translational control. The specificity of AMN082 to mGluR7 was confirmed using cultures made from *mGluR7* KO mice as no significant effects on protein synthesis were observed (Fig. 1E). Interestingly, MMPIP does not have any effects on protein synthesis, suggesting the possibility that basal mGluR7 activity might be low in cultured neurons. Because *mGluR7* KO mainly impacts the expression of mGluR7a isoform (Fig. 1E), our results suggest that activation of mGluR7, at least through mGluR7a, leads to *Fmr1*-independent repression of protein synthesis.

### Activation of mGluR7 represses protein synthesis via ERK1/2 and eIF4E signaling

We next sought to understand the signaling pathway by which activation of mGluR7 represses protein synthesis. It is known that mGluR7 is coupled with inhibitory G-protein (G$_i$) whose activation inhibits adenylyl cyclase and reduces cytosolic cAMP levels (Mitsukawa et al, 2005). We investigated two main regulators of activity-dependent protein synthesis in neurons that are influenced by the changes in cytosolic cAMP levels: (1) ERK1/2 and (2) mammalian target of rapamycin (mTOR) (Xie et al, 2011; Kim et al, 2010) To test whether one or both proteins are involved, we treated WT and *Fmr1* KO cortical neuron cultures at days-in-vitro (DIV) 14 with AMN082 (1 μM) for 2 h and then measured the levels of phosphorylated ERK1/2 (Fig. 1F) and phosphorylated mTOR (Fig. 1G) by western blotting. As shown, we observed significantly reduced levels of phosphorylated ERK1/2, but not phosphorylated mTOR, in both WT and *Fmr1* KO neurons.

It has been shown that phosphorylated ERK1/2 interacts with and phosphorylates mitogen-activated protein kinases-interacting kinases 1/2 (MNK1/2), which subsequently phosphorylates eukaryotic translation initiation factor 4E (eIF4E), leading to facilitated translation initiation (Waskiewicz et al, 1997; Pyronnet et al, 1999; Joshi et al, 1995) A recent study suggested that inhibition of ERK1/2 causes significant upregulation in phosphorylation of eukaryotic initiation factor 2-alpha (eIF2α) and inhibits cap-dependent and -independent translation (Parveen et al, 2021). To determine whether eIF4E, eIF2α, or both are impacted following activation of mGluR7, we measured the phosphorylation of eIF4E at Ser-209 and eIF2α at Ser-51 in WT and *Fmr1* KO cortical neuron cultures after treatment of AMN082 for 2 h. As shown, we found that eIF4E phosphorylation was significantly reduced (Fig. 1H), while eIF2α phosphorylation was not altered (Fig. 1I) after treatments with AMN082 in both WT and *Fmr1* KO neurons. Although we did not observe basally elevated eIF4E phosphorylation in *Fmr1* KO cultures, which is known to be a brain region- and age-dependent effect (Liu et al, 2022), our data suggest that AMN082 represses protein synthesis through eIF4E phosphorylation. To further test the suppressive effect of AMN082 on eIF4E, we

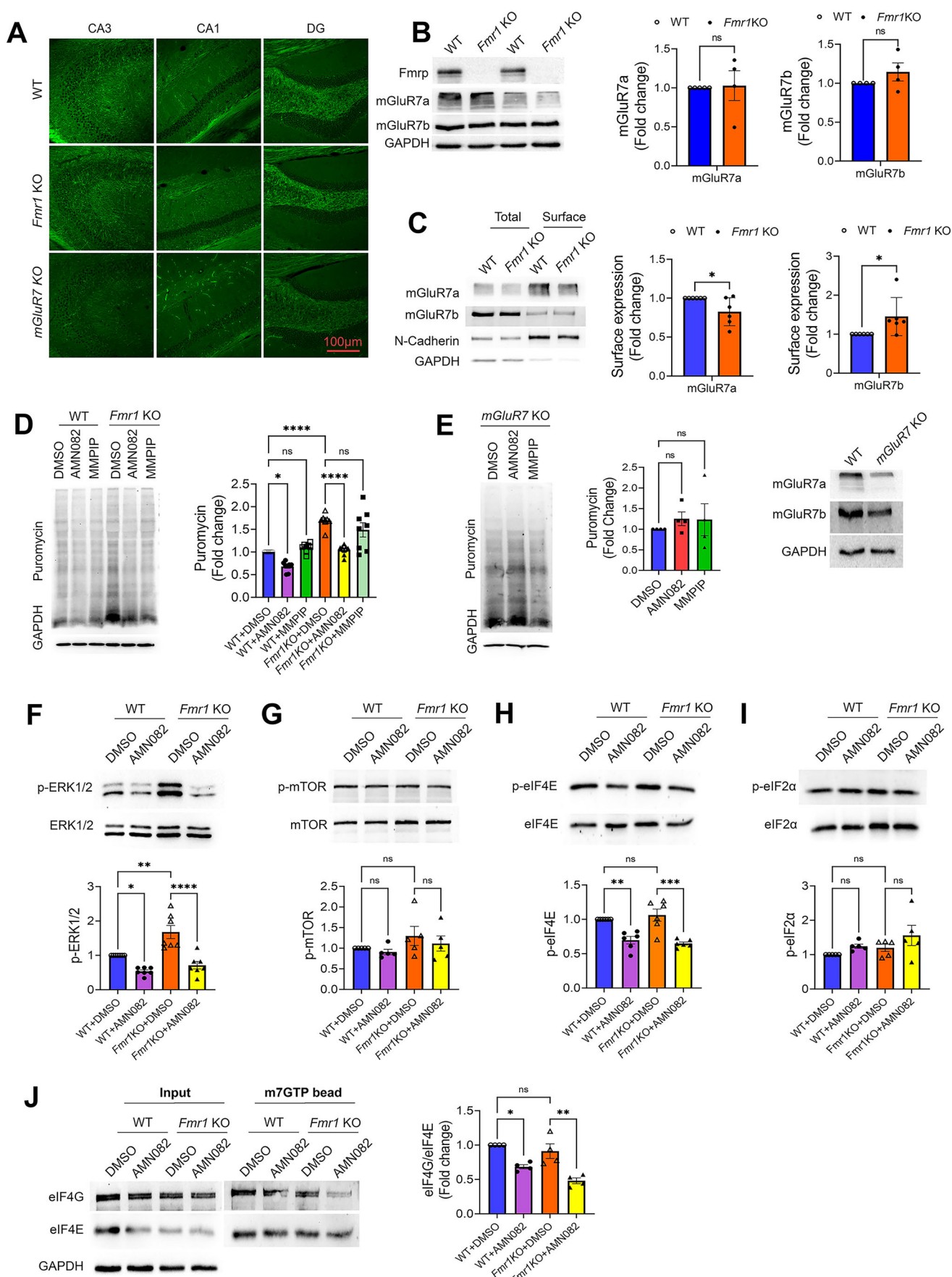

◄ Figure 1. Activation of mGluR7 reduces protein synthesis in both WT and *Fmr1* KO neurons.

(A) Representative fluorescence images showing expression of mGluR7a in CA1, CA3, and dentate gyrus (DG) in brain sections obtained from WT, *Fmr1* KO, and *mGluR7* KO mice at postnatal (P) day 60. (B) Representative western blots and quantification showing the expression of mGluR7a and mGluR7b in total brain lysate from WT and *Fmr1* KO mice at P60 (n = 4 mice; p = 0.8946 and 0.3038 for mGluR7a and mGluR7b, respectively). (C) Representative western blots and quantification of mGluR7a, mGluR7b, and surface protein marker N-Cadherin from total cell lysates or extracted surface protein fractions in primary cortical neuron cultures made from WT or *Fmr1* KO mice (n = 6 independent cultures; p = 0.0387 and 0.0463 for mGluR7a and mGluR7b, respectively). (D) Representative western blots and quantifications of puromycin and GAPDH from WT and *Fmr1* KO cortical neuron cultures at DIV 12–14 treated with DMSO, AMN082 (1 μM), or MMPIP (1 μM) for 2 h followed by treatment of puromycin (10 μg/ml) to label newly synthesized protein for another hour. (n = 7 and 6 for WT and *Fmr1* KO, respectively) (WT + DMSO vs WT + AMN082, p = 0.0285; WT + DMSO vs WT + MMPIP, p = 0.9141; WT + DMSO vs *Fmr1*KO + DMSO, p < 0.0001; *Fmr1*KO + DMSO vs *Fmr1*KO + AMN082, p < 0.0001; *Fmr1*KO + DMSO vs *Fmr1*KO + MMPIP, p = 0.3435). (E) Representative western blots and quantifications of puromycin and GAPDH from *mGluR7* KO cortical neuron cultures at DIV 12–14 treated with DMSO, AMN082 (1 μM), or MMPIP (1 μM) for 2 h followed by treatment of puromycin (10 μg/ml). A set of blots showing the levels of mGluR7a and mGluR7b in WT and *mGluR7* KO cultures is on the right (n = 4) (DMSO vs AMN082, p = 0.732; DMSO vs MMPIP, p = 0.6935). (F–I) Representative western blots and quantifications of ERK1/2 phosphorylation, mTOR phosphorylation, eIF4E phosphorylation and eIF2α phosphorylation from WT and *Fmr1* KO cortical neuron cultures at DIV 12–14 treated with DMSO or AMN082 (1 μM) for 2 h. For the quantification, phosphorylation signal of a specific protein was normalized to its total protein signal. (n = 3–5 independent cultures) (F) WT + DMSO vs WT + AMN082, p = 0.0339; WT + DMSO vs *Fmr1*KO + DMSO, p = 0.0017; *Fmr1*KO + DMSO vs *Fmr1*KO + AMN082, p < 0.0001. (G) WT + DMSO vs WT + AMN082, p = 0.9571; WT + DMSO vs *Fmr1*KO + DMSO, p = 0.4032; *Fmr1*KO + DMSO vs *Fmr1*KO + AMN082, p = 0.7514. (H) WT + DMSO vs WT + AMN082, p = 0.0074; WT + DMSO vs *Fmr1*KO + DMSO, p = 0.8646; *Fmr1*KO + DMSO vs *Fmr1*KO + AMN082, p = 0.0005. (I) WT + DMSO vs WT + AMN082, p = 0.7174; WT + DMSO vs *Fmr1*KO + DMSO, p = 0.8258; *Fmr1*KO + DMSO vs *Fmr1*KO + AMN082, p = 0.4377). (J) Representative western blots and quantification of eIF4E and eIF4G pulled down by m7GTP beads from WT and *Fmr1* KO cortical neuron cultures treated with DMSO or AMN082 (1 μM) at DIV 12–14 (n = 4 independent cultures) (WT + DMSO vs WT + AMN082, p = 0.0249; WT + DMSO vs *Fmr1*KO + DMSO, p = 0.7503; *Fmr1*KO + DMSO vs *Fmr1*KO + AMN082, p = 0.0038). Data Information: Data were analyzed by Student's t-test (B, C), one-way ANOVA (E), or two-way ANOVA (D, F–J) with Tukey test and presented as mean ± SEM with *p < 0.05, **p < 0.01, ***p < 0.001, ****p < 0.0001 and NS: non-significant. Source data are available online for this figure.

measured the ability of eIF4E to bind to the scaffolding protein eIF4G in cap-dependent translation initiation. To this end, we assessed eIF4E-eIF4G complex by m7GTP pull-down assay, as performed previously (Santini et al, 2017), in WT and *Fmr1* KO neuronal cultures treated with DMSO or AMN082. As shown (Fig. 1J), both WT and *Fmr1* KO cultures treated with AMN082 showed a significant decrease in the eIF4E-eIF4G interaction. These results confirmed the repressive effect of AMN082 on eIF4E signaling.

To validate our observation in vivo, we intraperitoneally injected male WT and *Fmr1* KO mice at 6–8 weeks of age with AMN082 (1 mg/kg) and puromycin (200 mg/kg) for 1 h to assess protein synthesis in the hippocampus. AMN082 is known to rapidly cross the blood-brain barrier (Mitsukawa et al, 2005). As shown (Fig. 2A), we observed a significant increase in protein synthesis in saline-treated *Fmr1* KO mice when compared to saline-treated WT mice, suggesting increased protein synthesis in the hippocampus of *Fmr1* KO mice. Importantly, treatment of AMN082 reduced protein synthesis in both WT and *Fmr1* KO mice, similar to what we observed in cultured neurons (Fig. 1D). These effects on protein synthesis can also be seen when we tested one of FMRP's target genes, protocadherin-7 (Pcdh7) in the hippocampus of WT or *Fmr1* KO mice injected with saline or AMN082 (Fig. EV1). The specificity of AMN082 to mGluR7 was confirmed as no significant effect was observed in *mGluR7* KO mice (Fig. 2B). We further tested ERK1/2 and eIF4E signaling in vivo and confirmed a decrease in phosphorylated ERK1/2 and eIF4E in both WT and Fmr1 KO mice treated with AMN082 (1 mg/kg) (Fig. 2C,D). Taken together, our results suggest that mGluR7 potentially acts through ERK1/2 and eIF4E to repress protein synthesis (Fig. 2E).

## Activation of mGluR7 reduces neuronal excitability

Activation of mGluR7 is known to lead to a reduction in presynaptic glutamate release via inhibition of P/Q-type Ca2+ channels (Martín et al, 2007) and reduction in neuronal excitability via inhibition of N-type

calcium channels (Millán et al, 2002). However, it is unclear whether mGluR7 activation is capable of attenuating pathological hyperexcitability, which is one of the key neuronal abnormalities in FXS. To begin testing this possibility, we first tested the effect of AMN082 on neuronal network activity using a multielectrode array (MEA) recording system (Maestro Edge, Axion Biosystems). We treated WT and *Fmr1* KO neurons with dimethyl sulfoxide (DMSO), AMN082 (1 μM), or the selective mGluR7 antagonist, MMPIP (1 μM) for 2 h and compared network activity pre- and post-treatment. As shown in Fig. 3A, a drastic reduction in network activity can be seen with a corresponding reduction in spontaneous spike rate, burst duration, and burst frequency in both WT and *Fmr1* KO cultures. In a manner similar to that observed regarding protein synthesis (Fig. 1D), MMPIP did not elicit any significant effects on network activity in either WT or *Fmr1* KO cultures.

The findings from the MEA recordings prompted us to evaluate the effect of mGluR7 activation at a single cell level. We performed whole-cell patch-clamp recording in WT and *Fmr1* KO cortical neurons at DIV 14. We used a current-clamp recording to measure the action potential firing rate after delivering constant somatic current pulses for durations of 500 ms in the range of 0 to 200 pA (Liu et al, 2021). As anticipated, mGluR7 activation caused a significant reduction in the action potential firing rate in both WT and *Fmr1* KO neurons (Fig. 3B). Together, our results from MEA and whole-cell patch-clamp recordings indicate that activation of mGluR7 causes a decrease in neuronal network activity and excitability to a similar degree in both WT and *Fmr1* KO cultures.

FXS patients and animal models all exhibit elevated circuit excitability with seizures as a common comorbidity. The audiogenic seizure (AGS) test is commonly used to measure circuit hyperexcitability in *Fmr1* KO mice (Ronesi et al, 2012), and we aimed to determine whether activation of mGluR7 can reduce the susceptibility to AGS in *Fmr1* KO mice. We intraperitoneally injected 3-week-old *Fmr1* KO mice with saline or AMN082 (1 mg/kg). WT mice were excluded from this experiment because they typically do not show audiogenic seizures (Guo et al, 2016). Thirty minutes after the injection, the mice were presented with 110 dB

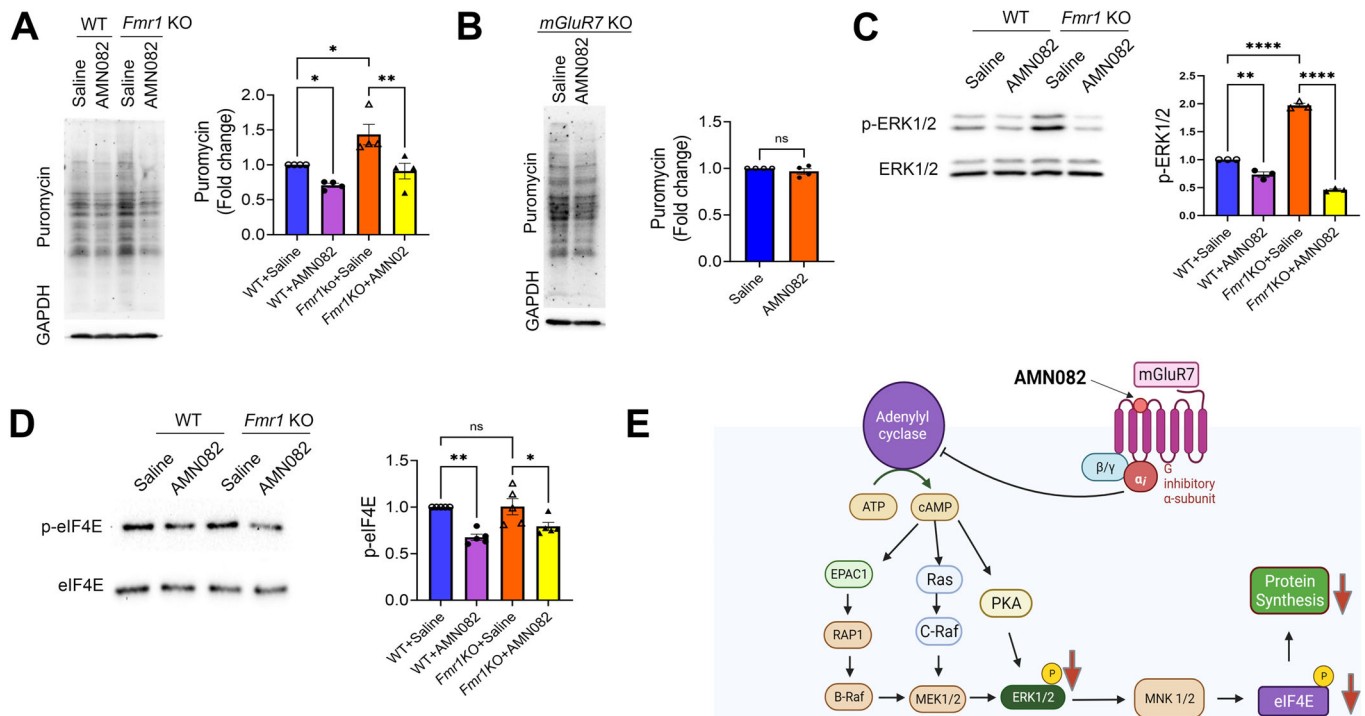

**Figure 2. mGluR7 activation represses protein synthesis and phosphorylation of ERK1/2 and eIF4E in WT and *Fmr1* KO mouse hippocampus.**

(A) Representative western blots (left) and quantification (right) of puromycin and GAPDH in the hippocampus of 6–8-week-old WT and *Fmr1* KO mice injected with saline or AMN082 (1 mg/kg) and puromycin (200 mg/kg) for 1 h. ($n = 4$ mice per treatment group) (WT+Saline vs WT + AMN082, $p = 0.0463$; WT+Saline vs *Fmr1*KO +Saline, $p = 0.0135$; *Fmr1*KO+Saline vs *Fmr1*KO + AMN082, $p = 0.0042$). (B) Representative western blots of puromycin and GAPDH in the hippocampus of *mGluR7* KO mice injected with saline or AMN082 (1 mg/kg) and puromycin (200 mg/kg) for 1 h ($n = 4$ mice per treatment group; $p = 0.3206$). (C, D) Left: representative western blots of p-ERK1/2 and p-eIF4E in hippocampal lysates from WT and *Fmr1* KO mice treated with AMN082 (1 mg/kg) for 1 h and the quantification (on right) showing phosphorylated protein levels normalized to their respective total protein levels. ($n = 5$ mice) (C: WT+Saline vs WT + AMN082, $p = 0.0044$; WT+Saline vs *Fmr1*KO +Saline, $p < 0.0001$; *Fmr1*KO+Saline vs *Fmr1*KO + AMN082, $p < 0.0001$. (D) WT+Saline vs WT + AMN082, $p = 0.0028$; WT+Saline vs *Fmr1*KO+Saline, $p = 0.9998$; *Fmr1*KO+Saline vs *Fmr1*KO + AMN082, $p = 0.0451$). (E) The schematic showing the signaling pathway implicated in the reduction of protein synthesis following the activation of mGluR7. Data Information: Data were analyzed by Student's *t*-test (B) or two-way ANOVA with Tukey test (A, C, D) and presented as mean ± SEM with *$p < 0.05$, **$p < 0.01$, ****$p < 0.0001$ and NS: non-significant. Source data are available online for this figure.

sound via a personal alarm for 2 min while seizure behavior was scored (Fig. 3C top). As shown (Fig. 3C bottom), treatment with AMN082 led to significant reduction in the susceptibility to AGS in *Fmr1* KO mice. These results suggest that activation of mGluR7 can alleviate pathological hyperexcitability in *Fmr1* KO mice.

## Activation of mGluR7 reduces autism-like behavior in *Fmr1* KO mice

Neural circuit hyperexcitability has been linked to other behavioral abnormalities in *Fmr1* KO mice, such as repetitive behaviors (Hussein et al, 2023). Because activation of mGluR7 can lead to a reduction in neuronal hyperexcitability in *Fmr1* KO neurons, we used the marble burying test to assess the effect of mGluR7 on repetitive behavior in *Fmr1* KO mice. As shown in Fig. 4A, saline-injected *Fmr1* KO mice buried significantly more marbles than saline-injected WT mice, suggesting an elevation in repetitive behavior in *Fmr1* KO mice. Importantly, injection of AMN082 (1 mg/kg) for 1 h corrects such a behavior in *Fmr1* KO mice but has no effects in WT mice (Fig. 4A). This finding suggests that activation of mGluR7 leads to an efficient reduction in repetitive behavior in *Fmr1* KO mice.

Because a marble burying test can be influenced by the animal's locomotion and anxiety behavior, we used an open field test to assess these behaviors. Following treatment with saline or AMN082, mice were allowed to explore an open field arena (67 cm x 67 cm) for 5 min. As shown in Fig. 4B, based on measurements of the distance traveled, immobile time, and time in the center zone, we did not observe any significant changes in WT or *Fmr1* KO mice treated with either saline or AMN082. The saline-injected *Fmr1* KO mice stayed slightly longer in the center zone, but such an effect was not affected by AMN082 treatment. Together, we conclude that AMN082 does not affect locomotion or anxiety behavior in WT or *Fmr1* KO mice.

Another commonly observed behavioral abnormality in FXS is impaired social interaction (Gkogkas et al, 2014; Mineur et al, 2006; Fyke et al, 2018). We used a three-chamber social interaction test to examine whether mGluR7 activation has an impact on social interaction. The test was divided into two sessions. In the first session, the mouse was allowed to freely explore and habituate in the arena. The second session was designed to test the sociability of the test mouse with a stranger mouse (Fig. 4C). Sociability was measured by calculating the time the test mouse spent interacting with the stranger mouse as opposed to the empty cage.

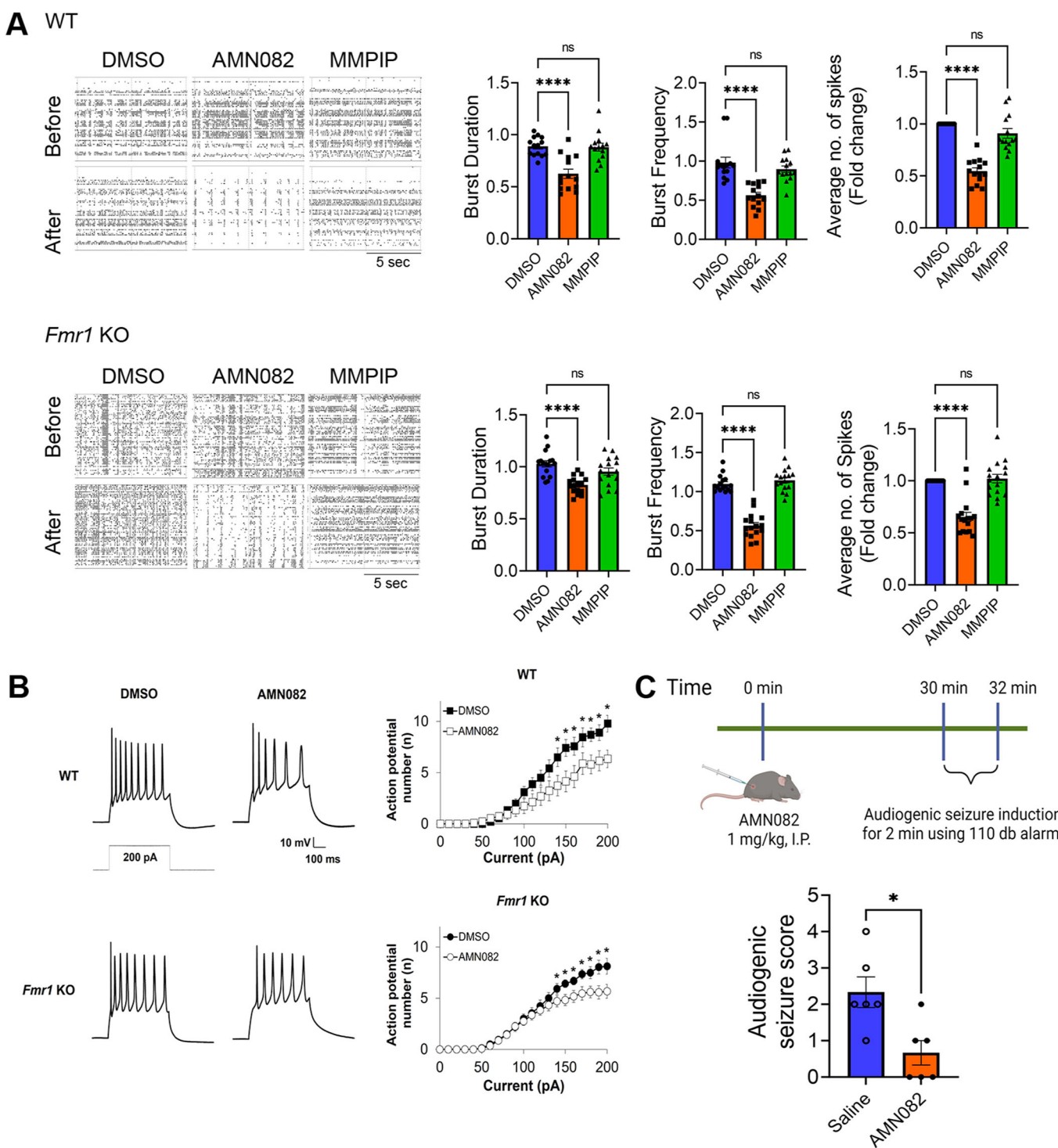

Interestingly, both WT and *Fmr1* KO mice showed similar sociability with the stranger mouse, and AMN082 did not produce significant effects. It is worth noting that studies in the field have reported inconsistent observation regarding social interaction, locomotion, or anxiety in *Fmr1* KO mice (Saré et al, 2016; Eadie et al, 2009). These discrepancies are likely caused by different genetic background of the mice and different testing environment employed in each study. In conclusion, our data suggests that activation of mGluR7 alleviates repetitive behavior without affecting locomotion, anxiety, or sociability in *Fmr1* KO mice.

## Activation of mGluR7 improves learning and memory in *Fmr1* KO mice

Because FXS is associated with intellectual disability, we next investigated the effects of mGluR7 activation on learning and

Figure 3. Activation of mGluR7 reduces neuronal excitability and susceptibility to audiogenic seizure in *Fmr1* KO mice.

(A) Representative raster plots of spontaneous spikes from WT (top) and *Fmr1* (bottom) cortical neuron cultures treated with DMSO, AMN082 (1 μM), or MMPIP (1 μM) for 2 h at DIV 12–14. Quantification of burst duration, burst frequency and average number of spikes by comparing "after treatment" to "before treatment", from the same culture was shown on the right (*n* = 5 independent cultures). (WT, Burst Duration: DMSO vs AMN082, *p* < 0.0001; DMSO vs MMPIP, *p* = 0.9924; Burst Frequency: DMSO vs AMN082, *p* < 0.0001; DMSO vs MMPIP, *p* = 0.4762; Average number of spikes: DMSO vs AMN082, *p* < 0.0001; DMSO vs MMPIP, *p* = 0.1294. *Fmr1* KO: Burst Duration: DMSO vs AMN082, *p* < 0.0001; DMSO vs MMPIP, *p* = 0.1751; Burst Frequency: DMSO vs AMN082, *p* < 0.0001; DMSO vs MMPIP, *p* = 0.5922; Average number of spikes: DMSO vs AMN082, *p* < 0.0001; DMSO vs MMPIP, *p* = 0.8932). (B) Left: Representative traces of action potentials induced by 200 pA from wild-type (top) or *Fmr1* KO (bottom) cortical neurons treated with DMSO or AMN082 (1 μM) for 2 h. Right: Average action potential firing rates (Hz) evoked by 0–200 pA injection from wild-type (top) or *Fmr1* KO (bottom) neurons treated with DMSO or AMN082 (1 μM) (*n* = 12–14 neurons per treatment group) (WT: *p* values for current stimulations between 140 pA and 200 pA are 0.0477, 0.0335, 0.0438, 0.0424, 0.0359, 0.0317, and 0.0348. *Fmr1* KO: *p* values for current stimulations between 140 pA and 200 pA are 0.0314, 0.0319, 0.0494, 0.0287, 0.0450, 0.0245, and 0.0266). (C) A schematic showing the experimental design for audiogenic seizure in *Fmr1* KO mice (top). Quantification of seizure scores after *Fmr1* KO mice were injected with saline or AMN082 (1 mg/kg) for 30 min (bottom) (*n* = 6 per treatment group; *p* = 0.0281). Data Information: Data were analyzed by one-way ANOVA with Tukey test (A, B) or two-tailed Mann–Whitney test (C) and presented as mean ± SEM with **p* < 0.05, ****p* < 0.0001 and NS: non-significant. Source data are available online for this figure.

memory in *Fmr1* KO mice. We first employed a novel object recognition test to test object recognition memory. WT and *Fmr1* KO mice injected with saline or AMN082 (1 mg/kg) were allowed to explore two identical objects during the first session, followed by a second session where one of the two original objects was replaced with a novel object. The behavior was videotaped and analyzed by Animal Tracker software (Gulyás et al, 2016), and the preference for novel objects was calculated (Fig. 5A, top). In comparison to WT mice, *Fmr1* KO mice exhibited a reduction in recognition memory, demonstrated by a significant reduction in preference index. Injection of AMN082 for 1 h corrected this phenotype in *Fmr1* KO mice without significant effects in WT mice (Fig. 5A, bottom).

To further evaluate learning and memory, we applied a contextual fear conditioning test to evaluate associative learning and memory, which has been shown to be impaired in *Fmr1* KO mice (Ding et al, 2014). WT and *Fmr1* KO mice were injected with saline or AMN082 (1 mg/kg) for 1 h and subjected to the test as shown in the schematic paradigm (Fig. 5B, top). Each test session was videotaped, and the duration of freezing behavior was calculated as described earlier (Lee et al, 2021). Saline-treated *Fmr1* KO mice exhibited significantly reduced freezing behavior when compared with respective WT control mice. Notably, AMN082-treated *Fmr1* KO mice showed a significant increase in freezing behavior, while AMN082 did not induce significant effects in WT mice (Fig. 5B, bottom).

Because *Fmr1* KO mice have been shown to exhibit impaired spatial learning and memory (Baker et al, 2010) we aimed to evaluate the effects of AMN082 on spatial learning and memory using the Barnes maze test. During the test, the mouse was trained to locate the escape box placed under one of the holes to escape from the high-intensity light. Three trials (5 min each) were conducted each day for 4 days, and the average escape latency (time taken to locate and enter the escape box) and the number of errors made before locating the escape box were calculated. We did not observe any significant difference between WT and *Fmr1* KO mice in escaping behavior across all 4 days, and AMN082 treatment did not produce any further effects (Fig. 5C, top). On day 5, the mice were subject to a probe trial to assess spatial memory formation where the escape box was removed, and the duration of time spent in the target quadrant was measured. No significant difference was observed between WT and *Fmr1* KO mice, and AMN082 did not produce any effects in either genotype (Fig. 5C, bottom). These results suggest no spatial memory defects in this cohort of *Fmr1* KO

mice, which was similarly seen by other studies(Van Dam et al, 2000; Leach et al, 2016), and AMN082 did not lead to improvements in this behavior. Altogether, these data suggest that activation of mGluR7 produced some improvement in learning and memory based on our data from novel object recognition and contextual fear conditioning tests.

## Discussion

In this study, we found that activation of mGluR7 leads to a reduction in susceptibility to AGS, reduction in repetitive behavior, and improvement in learning and memory in *Fmr1* KO mice (Fig. 5D). Molecularly, we revealed repression of protein synthesis in an *Fmr1*-independent manner as a novel functional outcome of mGluR7 activation. This repression appears to take place via inhibition of ERK1/2 and eIF4E phosphorylation, which is a common signaling pathway involved in *de novo* protein synthesis (Waskiewicz et al, 1997; Pyronnet et al, 1999; Joshi et al, 1995). Because activation of mGluR7 is known to inhibit adenylyl cyclase, it is likely that reversed protein synthesis and behavior in *Fmr1* KO mice upon treatment of AMN082 are accompanied by reduction in cytosolic cAMP levels. This is consistent with the study by Sethna et al (Sethna et al, 2017) showing increased cAMP in *Fmr1* KO mouse model due to increased translation of type 1 adenylyl cyclase (Adcy1) mRNA, and its suppression improved behavioral abnormalities in *Fmr1* KO mouse model. However, other recent studies have shown lower cAMP levels in the human FXS cells as well as the mouse models of FXS (Berry-Kravis et al, 1995; Kelley et al, 2007; Berry-Kravis and Sklena, 1993). It is also reported that increasing cAMP levels using inhibitor of phosphodiesterase 4 (PDE4) significantly alleviated symptoms of FXS (Gurney et al, 2017; Berry-Kravis et al, 2021; Rosenheck et al, 2021). These contradicting studies suggest the complexity of molecular mechanism underlying behavioral abnormalities in *Fmr1* KO animal models. They also support the need for a future direction to cross-examine age-, brain region-, and cell type-specific effects observed in different studies.

Because activation of other glutamate receptors, such as N-methyl-D-aspartate (NMDA) receptors, mGluR1, and mGluR5, is known to promote protein synthesis, our findings suggest that mGluR7 activation potentially acts as a counterbalance to prevent overproduction of new proteins upon activity-associated stimulation. To test this possibility in the future, we may need to

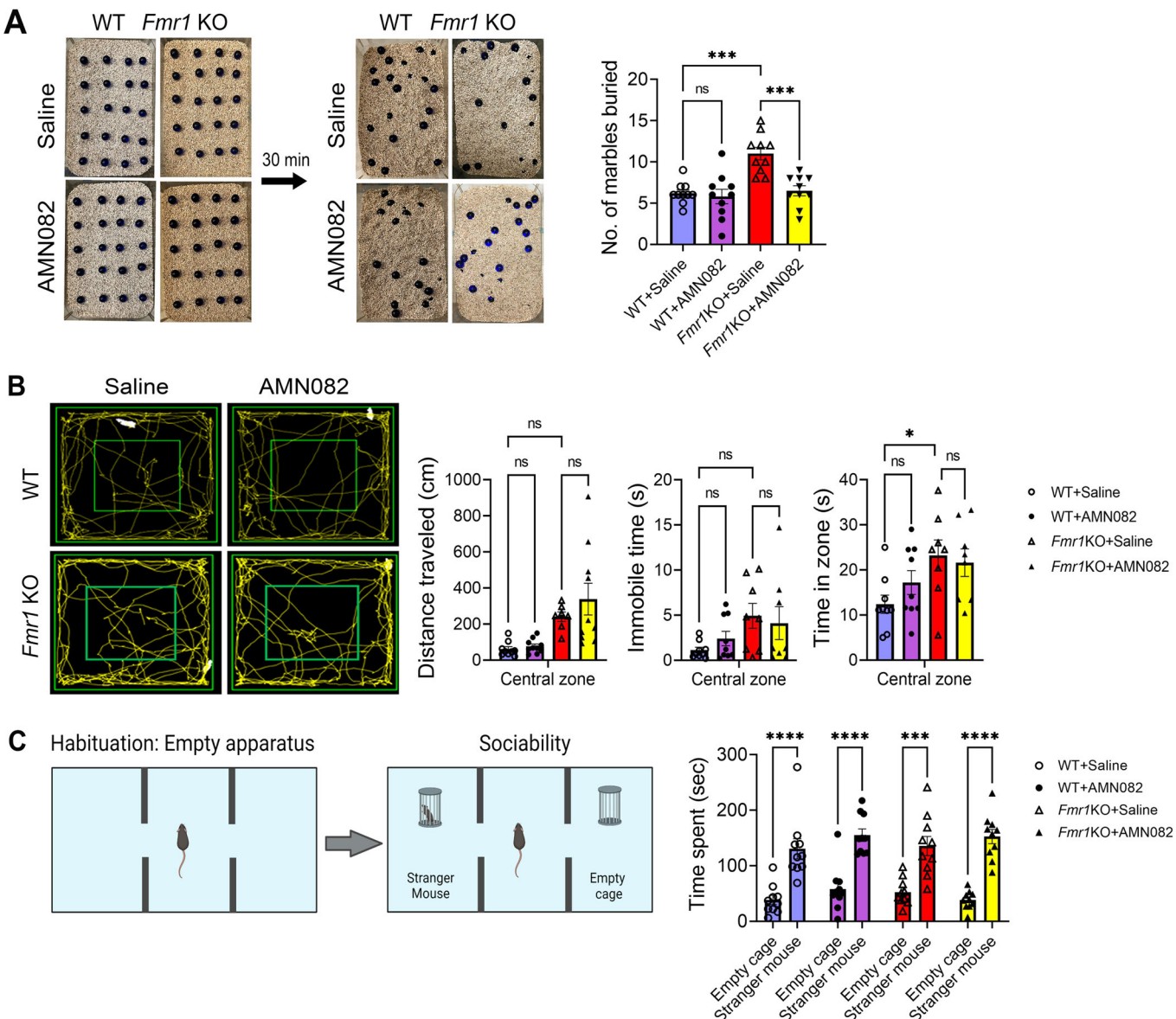

**Figure 4. Activation of mGluR7 ameliorates repetitive behavior without affecting locomotor activity or sociability in *Fmr1* KO mice.**

(A) Representative images of the marbles' configuration before and after allowing mice to bury marbles for 30 min (left). WT or *Fmr1* KO mice were intraperitoneally injected with saline or AMN082 (1 mg/kg) for 1 h before the test. Quantification of the marble burying activity is shown on the right ($n = 10$ mice per treatment group) (WT+Saline vs WT + AMN082, $p = 0.9791$; WT+Saline vs *Fmr1*KO+Saline, $p = 0.0004$; *Fmr1*KO+Saline vs *Fmr1*KO + AMN082, $p = 0.0008$). (B) Representative traces from a 5-min test period in an open-field arena (left) showing the movement of WT or *Fmr1* KO mice after intraperitoneally injected with saline or AMN082 (1 mg/kg) for 1 h. Quantification of total distance traveled, immobile time and time spent in central zone is shown on the right ($n = 10$ mice per treatment group) (Distance traveled: WT +Saline vs WT + AMN082, $p = 0.9961$; WT+Saline vs *Fmr1*KO+Saline, $p = 0.1124$; *Fmr1*KO+Saline vs *Fmr1*KO + AMN082, $p = 0.5765$. Immobile time: WT+Saline vs WT + AMN082, $p = 0.8525$; WT+Saline vs *Fmr1*KO+Saline, $p = 0.1211$; *Fmr1*KO+Saline vs *Fmr1*KO + AMN082, $p = 0.9636$. Time in zone: WT+Saline vs WT + AMN082, $p = 0.6004$; WT+Saline vs *Fmr1*KO+Saline, $p = 0.0482$; *Fmr1*KO+Saline vs *Fmr1*KO + AMN082, $p = 0.9786$). (C) The three-chamber social interaction test from WT or *Fmr1* KO mice after intraperitoneally injected with saline or AMN082 (1 mg/kg) for 1 h. Schematic representation depicting the sociability protocol for the three-chamber social interaction test was also shown ($n = 10$ mice per treatment group) (WT+Saline, $p < 0.0001$; WT + AMN082, $p < 0.0001$, *Fmr1*KO+Saline, $p = 0.0009$, and *Fmr1*KO + AMN082, $p < 0.0001$). Data Information: Data were analyzed by two-way ANOVA with Tukey's or Šidák's test and presented as mean ± SEM with \*$p < 0.05$, \*\*$p < 0.01$, \*\*\*$p < 0.001$ and NS: non-significant. Source data are available online for this figure.

understand how different glutamate receptors regulate protein synthesis in distinct directions. Does activity-dependent translational control occur via a biased manner where only certain types of glutamate receptors are activated to achieve either elevation or reduction of protein synthesis? Or does glutamate activate all

possible glutamate receptors, but the effect on protein synthesis depends on the availability of downstream signaling molecules that relay the information? Both scenarios may be true and could potentially happen simultaneously, and we propose to test them in a future study.

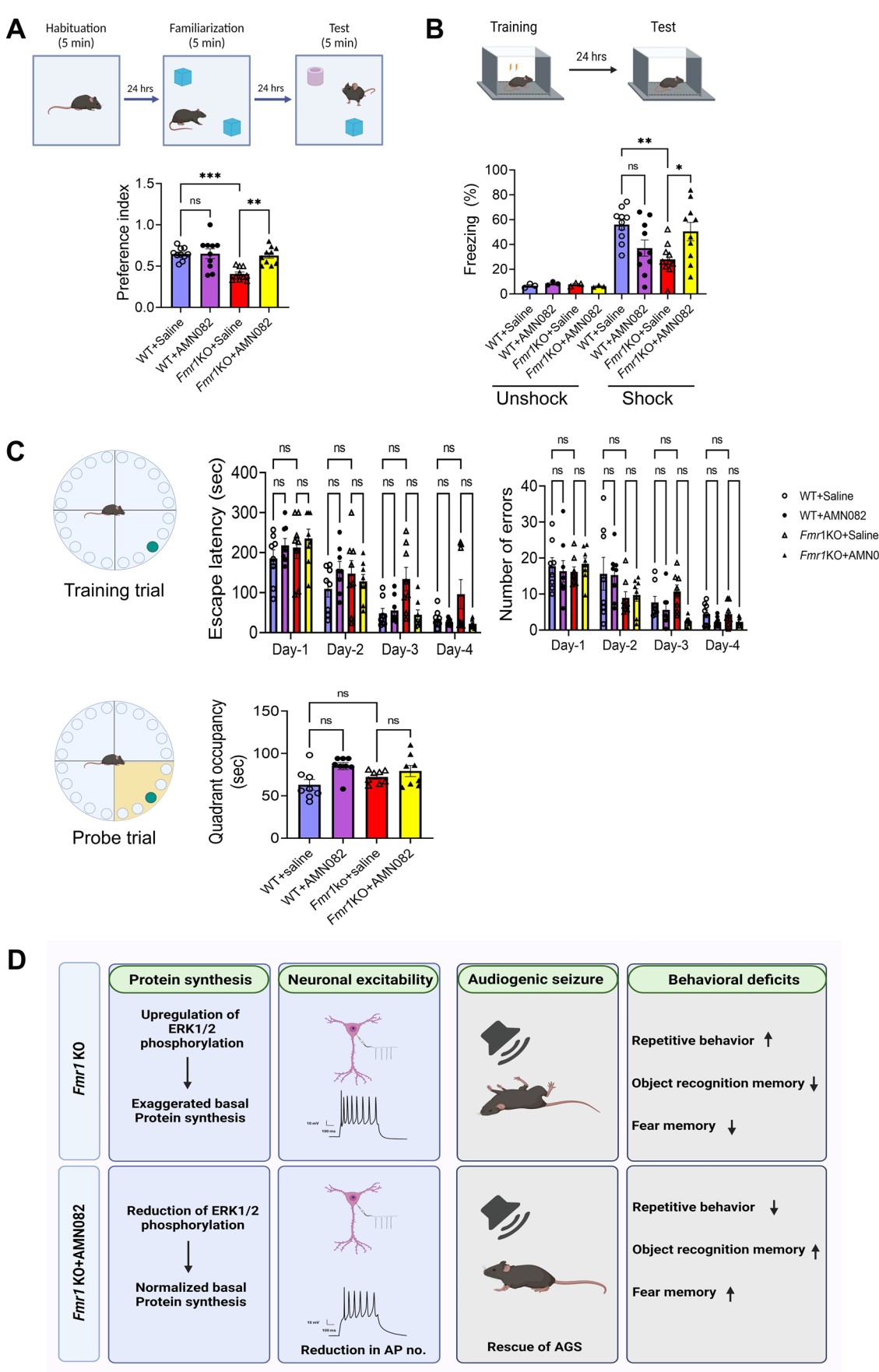

**Figure 5. Activation of mGluR7 improves learning and memory in _Fmr1_ KO mice.**

(A) A schematic showing the novel object recognition test paradigm (top) and the quantification of preference index from WT or _Fmr1_ KO mice after intraperitoneally injected with saline or AMN082 (1 mg/kg) for 1 h (bottom) ($n = 10$ mice per treatment group) (WT+Saline vs WT + AMN082, $p = 0.9999$; WT+Saline vs _Fmr1_KO +Saline, $p = 0.0008$; _Fmr1_KO+Saline vs _Fmr1_KO + AMN082, $p = 0.0002$). (B) A schematic depiction of contextual fear conditioning test paradigm (top) and quantification of the freezing behavior of WT or _Fmr1_ KO mice after intraperitoneally injected with saline or AMN082 (1 mg/kg) for 1 h (bottom) ($n = 10$ mice per treatment group) (In Shock groups: WT+Saline vs WT + AMN082, $p = 0.1642$; WT+Saline vs _Fmr1_KO+Saline, $p = 0.0066$; _Fmr1_KO+Saline vs _Fmr1_KO + AMN082, $p = 0.418$). (C) A schematic representation of Barnes maze test showing the training trail and probe trial. WT or _Fmr1_ KO mice intraperitoneally injected with saline or AMN082 (1 mg/kg) were trained for 4 days with 3 consecutive trials every day to locate the escape box (green circle). The escape latency and number of errors during the training trial were quantified. Quadrant occupancy on probe trial on day 5 was assessed by recording the time spent in the target area (yellow area) ($n = 10$ mice per treatment group) (Escape latency, Day 1: WT+Saline vs WT + AMN082, $p = 0.9999$; WT+Saline vs _Fmr1_KO+Saline, $p = 0.9999$; _Fmr1_KO+Saline vs _Fmr1_KO + AMN082, $p = 0.9999$; Day 2: WT+Saline vs WT + AMN082, $p = 0.9999$; WT+Saline vs _Fmr1_KO+Saline, $p = 0.9999$; _Fmr1_KO+Saline vs _Fmr1_KO + AMN082, $p = 0.9999$; Day 3: WT+Saline vs WT + AMN082, $p = 0.9999$; WT+Saline vs _Fmr1_KO+Saline, $p = 0.4132$; _Fmr1_KO+Saline vs _Fmr1_KO + AMN082, $p = 0.3069$; Day 1: WT+Saline vs WT + AMN082, $p = 0.9999$; WT+Saline vs _Fmr1_KO+Saline, $p = 0.9953$; _Fmr1_KO+Saline vs _Fmr1_KO + AMN082, $p = 0.7988$. Number of errors, Day 1: WT+Saline vs WT + AMN082, $p = 0.9999$; WT+Saline vs _Fmr1_KO+Saline, $p = 0.9999$; _Fmr1_KO+Saline vs _Fmr1_KO + AMN082, $p = 0.9999$; Day 2: WT+Saline vs WT + AMN082, $p = 0.9999$; WT+Saline vs _Fmr1_KO+Saline, $p = 0.8944$; _Fmr1_KO+Saline vs _Fmr1_KO + AMN082, $p = 0.9999$; Day 3: WT+Saline vs WT + AMN082, $p = 0.9999$; WT +Saline vs _Fmr1_KO+Saline, $p = 0.4095$; Day 4: WT+Saline vs WT + AMN082, $p = 0.9999$; WT+Saline vs _Fmr1_KO +Saline, $p = 0.9999$; _Fmr1_KO+Saline vs _Fmr1_KO + AMN082, $p = 0.9999$. Quadrant occupancy: WT+Saline vs WT + AMN082, $p = 0.0676$; WT+Saline vs _Fmr1_KO +Saline, $p = 0.8267$; _Fmr1_KO+Saline vs _Fmr1_KO + AMN082, $p = 0.9493$). Data Information: Data were analyzed by two-way ANOVA with Tukey's test and presented as mean ± SEM with *$p < 0.05$, **$p < 0.01$, ***$p < 0.001$ and NS: non-significant. (D) Summary of the effects from activation of mGluR7 on molecular and behavioral deficits in _Fmr1_ KO mice. Source data are available online for this figure.

Our data indicate that activation of mGluR7 can repress protein synthesis in both WT and _Fmr1_ KO neurons, suggesting that the effects are _Fmr1_-independent. This prediction is expected because no previous studies showing a dysregulation of mGluR7 in FXS patients or animal models have been reported, although our results show a slightly altered surface expression pattern of mGluR7a and mGluR7b in _Fmr1_ KO mice. We therefore conclude that while _Grm7_ is not a disease-causing gene, activation of mGluR7 is useful for correcting or alleviating the pathological defects in FXS. As we showed in this study, hyperexcitability, repetitive behavior, and memory deficits in _Fmr1_ KO mice were found to be significantly improved following activation of mGluR7. Given the many functions of FMRP and the extensive binding affinity toward numerous mRNAs by FMRP, the pathophysiology of FXS is extremely complex. We tested several known defects in _Fmr1_ KO mice, but it remains to be determined whether and how mGluR7 activation can correct other reported phenotypes in FXS or its animal models. For example, local protein synthesis in axons or dendrites especially for synaptic proteins is known to be impaired in _Fmr1_ KO mice (Daroles et al, 2016; Monday et al, 2022). It would be of particular interest to determine whether activation of mGluR7 can correct local protein synthesis in _Fmr1_ KO neurons. Another example would be to study whether and how AMN082 can rescue the excessive dendritic spines in _Fmr1_ KO neurons. Furthermore, since mGluR7 is expressed in both pre- and post-synaptic compartments (Palazzo et al, 2016), it will be important to study if the suppression of protein synthesis upon activation of mGluR7 is mediated via pre- and/or post- synaptic terminals. In addition, _Fmr1_ KO mice exhibit multiple translation-related defects in neural plasticity, including long-term potentiation (Tian et al, 2017), long-term depression (Niere et al, 2012), and homeostatic plasticity (Lee et al, 2018; Soden and Chen, 2010). Given that mGluR7 activation can repress protein synthesis in _Fmr1_ KO neurons, we expect that pharmacological activation of mGluR7 using AMN082 may improve, at least partially, one or more of these plasticity mechanisms. All such mechanisms would require substantial future efforts to validate. One major challenge remains unmet when targeting mGluR7 in vivo is the availability of a stable agonist. At present AMN082 is a highly potent, blood-brain-barrier permeable and commercially available mGluR7 agonist. However, its rapid breakdown in liver cells reduces its bioavailability (Sukoff Rizzo et al, 2011). On the other hand, the metabolites produced from breakdown of AMN082 are

shown to have an affinity for serotonin transporter SERT (Sukoff Rizzo et al, 2011). This issue limits chronic use of AMN082 in vivo and prompts another future direction to synthesize a rather more stable alternative of AMN082 that could improve the study of physiological functions of mGluR7 activation in vivo.

FXS is the most common cause of inherited autism and mGluR7 is encoded by an autism-linked gene. The rescue effect upon mGluR7 activation in _Fmr1_ KO mice suggests the possibility that AMN082 may exert beneficial effects toward other ASDs, particularly those with known defects in protein synthesis. For example, mutations or haploinsufficiency of phosphatase and tensin homolog (_PTEN_) results in autism (Butler et al, 2005) with elevated protein synthesis being one of the most common defects observed in disease animal models. PTEN-associated ASD also shares many similar phenotypes with FXS, including sensory hypersensitivity and seizures (Smith et al, 2016). Another example is tuberous sclerosis complex, which shares similar defects in protein synthesis as PTEN-associated ASD in addition to exhibiting seizure phenotypes (Di Nardo et al, 2009). It would be of particular interest to examine whether activation of mGluR7 can have a broader impact on other ASDs to reduce the hyperexcitability phenotypes and perhaps also improve cognition. In summary, targeting mGluR7 has the potential to open new avenues for the study of activity-dependent neural plasticity and neurodevelopment with possibilities to introduce new approaches to produce improvements in the quality of life of individuals affected by ASD. More research is needed and expected to broaden our understanding of mGluR7, which is strongly associated with autism but remains poorly understood.

## Methods

### Animals

All experiments using animals followed the guidelines of Animal Care and Use provided by the University of Illinois Institutional Animal Care and Use Committee (IACUC) and the guidelines of the Euthanasia of Animals provided by the American Veterinary

Medical Association (AVMA) to minimize animal suffering and the number of animals used. This study was performed under an approved IACUC animal protocol of University of Illinois at Urbana-Champaign (#20049 and #23016 to N.-P. Tsai.). We obtained WT (stock No. 00664) and *Fmr1* KO mice (stock No. 003025) from Jackson laboratory and *Grm7* KO mice from MMRRC (B6.129P2-Grm7tm1Dgen/Mmnc, stock No. 011626-UNC). Because of the higher prevalence of FXS in males, we only employed male mice in our study. Mice were housed in individually ventilated cages in 12 h light/dark cycle with *ad libitum* access to pelleted food and water. Although experiments were not performed using littermate mice, WT and *Fmr1* KO littermates were used to generate WT and *Fmr1* KO breeding cages. *Grm7* KO mice were identified by PCR with genomic DNA prepared from toe clips. For genotyping *Grm7* deleting allele, we used primers that detect *LacZ* allele to reflect the transgene: 5′-CGATCGTAATCACCC-GAGTGT-3′, and 5′-CCGTGGCCTGACTCATTCC-3′. The primers used to differentiate *wild-type* allele are 5′-GCGGATCCTGGACACTTGTT-3′, and 5′-GCGCCTGGAC-GAAAGTGA. For genotyping *Fmr1* deleting allele, a set of three primers were used: 5′-CACGAGACTAGTGAGACGTG-3′ (*Fmr1* KO), 5′-CACGAGACTAGTGAGACGTG-3′ (*wild-type*), and 5′-CTTCTGGCACCTCCAGCTT-3′ (common reverse primer).

### Primary neuronal culture

The primary cortical neuronal culture was performed as previously described using mice at post-natal day 0–1 (Tsai et al, 2012). Cortices were dissected and incubated with trypsin for 8–10 min at 37 °C. Next, trypsin was neutralized by the addition of fetal bovine serum (FBS) supplemented HBSS and washed twice with pre-warm HBSS. Cortices were then briefly homogenized in complete DMEM and plated on poly-D-lysine (0.05 mg/ml) coated 6-well plates. After 3–5 h, the medium was replaced with Neurobasal A medium (10888022, ThermoFisher Scientific) supplemented with 2 mM Glutamax (35050061, Invitrogen), B27 supplement (17504001, Invitrogen) and 1 μM Ara-C ((β-D-Arabinofuranosyl) cytosine) (C1768, Sigma-Aldrich). Cultures were maintained at 37 °C with 5% $CO_2$. Half of the medium was changed on days-in-vitro (DIV) 2 and thereafter every 3–4 days. Experiments were performed when cultures were at DIV 12–16.

### Reagents

Dimethyl sulfoxide was from Thermo Fisher Scientific (#BP231). Antibodies for western blotting were purchased from Sigma: rabbit anti-mGluR7a (#07-239, 1:2000 dilution) and mouse anti-puromycin (#MABE343, 1:1000 dilution); from Synaptic Systems: rabbit anti-mGluR7b (#191 203, 1:2000 dilution); from Proteintech: anti-GAPDH (#60004-1, 1:2500); from Thermo Fisher Scientific: rabbit anti-phospho-eIF2α (#MA5-15133, 1:1000 dilution); from Bioss: rabbit anti-PCDH7 (#bs-11085R, 1:1000 dilution); from Abcam: anti-phospho-eIF4E (#2069, 1:1000 dilution); from Cell Signaling Technology: rabbit anti-FMRP (#7104, 1:2500 dilution); rabbit anti-N-cadherin (#13116, 1:2500 dilution); rabbit anti-ERK1/2 (#4695, 1:5000 dilution); rabbit anti-phospho-ERK1/2 (#4370, 1:5000 dilution); rabbit anti-mTOR (#2972, 1:2500 dilution), rabbit anti-phospho-mTOR (#2971, 1:2500 dilution); rabbit anti-eIF4E (#2067, 1:2500 dilution); rabbit anti-eIF2α (#5324, 1:2500 dilution); rabbit anti-eIF4G (#2498, 1:2500 dilution). Secondary antibodies were from Cell Signaling Technology: anti-mouse HRP (#7076, 1:2500 dilution) and from Jackson ImmunoResearch: anti-rabbit HRP (#711-035-152, 1:2500 dilution).

### Western blotting

Tissue or cell cultures were lysed in ice-cold lysis buffer (137 mM NaCl, 20 mM Tris-HCL, 2 mM EDTA, and 1% Triton X-100, pH 8.0) supplemented with protease inhibitors (A32963, ThermoFisher Scientific) and phosphatase inhibitors (P2850; Sigma-Aldrich). Lysates were briefly sonicated and centrifuged. Supernatants were collected and protein concentration was measured using Bradford's method. SDS buffer (40% glycerol; 240 mM Tris-HCl, pH 6.8; 8% sodium dodecyl sulfate; 0.04% bromophenol blue; and 5% β-mercaptoethanol) was added to the lysates and heated at 95 °C for 10 min. Samples were then separated on SDS-PAGE gel and transferred onto a PVDF membrane (sc-3723, Santa Cruz Biotechnology). Membranes were blocked with 1% bovine serum albumin (BSA, BP9706100, ThermoFisher Scientific) in Tris-buffered saline Tween-20 buffer (TBST; [20 mM Tris, pH 7.5; 150 mM NaCl; 0.1% Tween-20]) for 30 min. Subsequently, membranes were incubated with primary antibodies overnight at 4 °C. Next, the membranes were washed 3 times in TBST and incubated with HRP-conjugated secondary antibody in 5% non-fat skimmed milk in TBST for an hour at 25 °C. Membranes were washed with TBST for 3 times and developed by using an enhanced chemiluminescence reagent and detected by an iBright imaging system (ThermoFisher Scientific, Waltham, MA). Band of the protein of interest were analyzed by ImageJ software (National Institute of Health).

### Surface protein biotinylation

For surface biotinylation, primary cortical neurons were plated at a density of $5 \times 10^5$ per well in 6-well plates as described previously (Nair et al, 2021). Cultures at DIV 14–16 were incubated on ice for 10 min followed by washing twice with DPBS (Dulbecco's phosphate-buffered saline, ThermoFisher Scientific 14200-059). Cultures were then biotin-labeled by incubating with 0.3 mg/ml Sulfo-NHS-SS-biotin (21331, ThermoFisher Scientific) solution for 10 min. Unbound biotin was scavenged by adding 100 mM $NH_4Cl$ followed by three washes with DPBS. Biotin-labeled cultures were then lysed in ice-cold lysis buffer and sonicated briefly. Lysates were centrifuged and supernatants were incubated with Streptavidin-Agarose beads (S1638, Merck) for an hour at 4 °C. After incubation, the lysates were removed by centrifugation and pelleted Streptavidin-Agarose beads were incubated with 4x sample buffer at 95 °C for 10 min to elute biotin-labeled surface proteins from beads. Eluted biotin-labeled surface protein was processed for western blotting as described above.

### m7GTP pull-down assay

m7GTP pull-down assay was performed in WT and *Fmr1* KO primary neuron cultures. The assay was performed as previously described (Santini et al, 2017). In brief, following treatment of DMSO or AMN082, neurons were harvested and sonicated in ice-

cold lysis buffer (137 mM NaCl, 20 mM Tris-HCL, 2 mM EDTA, and 1% Triton X-100, pH 8.0) supplemented with protease inhibitors (A32963, ThermoFisher Scientific) and phosphatase inhibitors (P2850; Sigma-Aldrich). Two hundred μg protein from each treatment condition was incubated with 20 μl m7GTP beads (m7GTP-001A, Creative BioMart) on a rotating mixer for 2 h. Following incubation, beads were pelleted by centrifugation at 6000 rpm for 1 min. Beads were then washed three times using cold lysis buffer. The protein complexes bound by the m7GTP beads were eluted in 4X SDS buffer and subjected to western blotting.

## Immunohistochemistry and imaging

Mice were anesthetized using isoflurane inhalation and transcardially perfused with PBS containing 10 units/ml heparin sodium (411210010, ThermoFisher Scientific) followed by 4% paraformaldehyde (PFA). The brains harvested were stored in 4% PFA overnight and then transferred to 10, 20, and 30% sucrose solution every 24 h at 4 °C. Brains were then cryosectioned in Leica 3050S cryotome and 15 μm sections were obtained. For immunostaining, sections were placed on gelatin-coated slides and incubated in antigen unmasking solution (H-3300, Vector Labs) at 70 °C for 40 min in a water bath. Sections were washed 3 times with PBS for 5 min each and permeabilized with 0.3% Triton X-100 for 10 min and then blocked with blocking buffer (1% bovine serum albumin, 3% normal goat serum and 0.3% Triton X-100) for 1 h at 25 °C. Sections were then probed with primary antibody prepared in blocking buffer and incubated overnight at 4 °C with anti-mGluR7a antibody (ab302530, Abcam, 1:250 dilution). Sections were washed with PBS and probed with Alexa488-conjugated goat anti-rabbit secondary antibody (BA-1000-1.5, Vector Labs) prepared in PBS at 1:1000 dilution and incubated for 2 h at 25 °C. Sections were washed with PBS and mounted using a mounting medium with DAPI (P36931, ThermoFisher Scientific). Imaging was performed in Zeiss LSM 700 Confocal microscope using 405 and 488 nm lasers. Images were acquired in z-stacks with 20X objective at 0.5X digital zoom and were processed using ImageJ software (National Institute of Health).

## MEA recording

Multielectrode array (MEA) recordings were performed using Maestro Edge (Axion Biosystems) with Cytoview MEA6 plates (6-well plates). Field potentials were recorded at each electrode relative to the ground electrode with a sampling frequency of 12.5 kHz. Followed by 30 min baseline recording (before), neurons were treated with indicated drugs for 2 h and recorded for another 30 min. To avoid the effect of change in physical movement on network activity, only the last 15 min of the recordings were used for data analysis. Axis Navigator version 3.3 software (Axion Biosystems) was used for spike extraction from raw electrical signals. After filtering the spike detector setting for each electrode was independently set at the threshold of ±6 standard deviation. Therefore, activity above the threshold was counted as a spike and included in data for analysis as previously described (Jewett et al, 2016). The total number of spikes was normalized to the number of electrodes in each well. The average number of spikes was calculated and expressed as fold changes with respect to the control. For detection of burst, a minimum of 5 spikes with a

maximum 100 ms spike interval was set for individual electrodes as described earlier (Jewett et al, 2016). Analysis of burst duration and burst frequency was performed using Axis Navigator version 3.3 software.

## Whole-cell patch-clamp recordings

Whole-cell patch-clamp recordings of action potential (AP) firing were carried out at 23–25 °C in a submersion chamber continuously perfused with ACSF containing (in mM): 119 NaCl, 2.5 KCl, 4 CaCl$_2$, 4 MgCl$_2$, 1 NaH$_2$PO$_4$, 26 NaHCO$_3$ and 11 D-Glucose, saturated with 95% O$_2$/5% CO$_2$ (pH 7.4, 310 mOsm), and were performed in the presence of fast synaptic transmission blockers: CNQX (20 μM); DL-APV (200 μM); and PTX (100 μM). Recording pipettes had a resistance of 4–6 MΩ when filled with an internal solution containing (in mM): 130 K-gluconate, 6 KCl, 3 NaCl, 10 HEPES, 0.2 EGTA, 4 Mg-ATP, 0.4 Na-GTP, 14 Tris-phosphocreatine (pH 7.25, 285 mOsm). Neurons were held at −60 mV. Action potential firing rates were measured upon delivering constant current pulses of 500 ms in the range 0–200 pA, and the number of action potentials was averaged from 3 to 5 individual sweeps for current intensity. Neurons were omitted if the resting membrane potential was ≥50 mV or if no action potentials were discharged. No series resistance compensation was used. The data were recorded using a Multiclamp 700B amplifier, Digidata 1550B, and the pClamp 10.6 (Molecular Devices). Recordings were filtered at 2 kHz and digitized at 10 kHz. Data analyses were performed using Clampfit 10.6 (Molecular Devices).

## In vivo puromycin labeling

Six- to eight-week-old WT, Fmr1 KO, and mGluR7KO mice were intraperitoneally injected with saline or AMN082 (1 mg/kg) and puromycin (200 mg/kg). One hour after the injections, mice were anesthetized using isoflurane inhalation and hippocampi were dissected out and flash-frozen in liquid N$_2$. Hippocampi were then lysed in ice-cold lysis buffer by sonication. Lysates were incubated with 4X SDS buffer at 95 °C for 10 min and subjected to western blotting.

## Audiogenic seizure assay

Fmr1 KO mice at post-natal day 20–22 were kept in a standard housing cage with minimum external noise to avoid auditory desensitization. Animals were intraperitoneally injected with saline or AMN082 (1 mg/kg). After 30 min, the mice were habituated in the transparent plastic box (28 × 17.5 × 12 cm) for 2 min before the onset of an auditory stimulus of 110 dB SPL (Personal alarm, Radioshack model 49–1010) for 2 min. The mice were videotaped during this time and scored for behavioral phenotype: 0 = no response, 1 = wild running, 2 = tonic-clonic seizures, 3 = status epilepticus, and 4 = death as described previously (Ronesi et al, 2012).

## Behavioral tests

All the behavioral tests were conducted on WT and Fmr1 KO mice at 6–8 weeks of age weighing 24–27 g unless otherwise specified. Mice were brought to the behavior testing room 30 min before the

test and housed in their home cages. The room was dimly lit at 50 lux and low background noise (~65 db). Behavioral apparatuses were thoroughly cleaned before and after every test session with 70% ethanol to avoid olfactory bias. Detailed procedures are provided below.

## Marble burying test

In a polycarbonate cage ($26 \times 48 \times 20$ cm) a total of 20 marbles were placed on the surface of bedding which was approximately 5 cm deep. Marbles were arranged in a $5 \times 4$ array. Mouse was introduced into the cage and allowed to remain inside for 30 min. Afterward, the mouse was taken out and a photograph of the cage floor was taken, and the total number of buried marbles was counted. A marble was classified as "buried" if two-thirds or more of it was concealed beneath the bedding. To carry out the next set of experiments, a clean cage filled with new bedding and marbles was used, following the same procedure as above.

## Open-field test

The open field test was carried out to evaluate the locomotion and anxiety behavior of mice. The mouse was placed in the center of a plexiglass box ($67 \times 67 \times 31$ cm) and allowed to freely explore the arena for 5 min. The movement of the mouse was recorded with an overhead camera and the video was analyzed using the Animal-Tracker plugin in ImageJ (National Institute of Health). The open-field arena was virtually divided into central and outer zones. The movement trajectory, velocity, immobile time, and total distance traveled by the test animal were calculated.

## Social interaction test

The social interaction test was conducted in a plexiglass chamber measuring $20 \times 40 \times 25$ cm and was divided into 3 parts by transparent walls with small openings to allow free movement of test animals between all three compartments. The test consisted of 2 sessions: habituation, and sociability sessions. Each session lasted for 10 min and was video recorded using an overhead camera. During the habituation session, the mouse was introduced to the middle chamber and allowed to explore all three chambers. For the sociability session, a stranger mouse of the same age and sex was placed in a wired cylinder and placed in the left chamber while an empty wired cylinder was placed in the right chamber. The test mouse was then reintroduced into the middle chamber and allowed to explore the left and the right chamber. The time spent by the test mouse to interact with the first stranger mouse over an empty cage was represented as sociability.

## Novel object recognition test

To test the object recognition memory, novel object recognition test was performed as previously described (Lee et al, 2021). On day-1, the mouse was placed in the empty testing chamber ($25 \times 25$ cm) and allowed to habituate for 5 min. On day-2, the mouse was allowed to freely explore two identical objects in the box before returning to the home cage. On day 3, one of the two identical objects was replaced with a novel object and the mouse was allowed to explore them for 10 min. The preference for exploring the novel

object was calculated by dividing the time elapsed by the mouse exploring the novel object ($T_{novel}$) by the time elapsed for exploring both familiar and novel objects ($T_{novel} + T_{familiar}$) and expressed as the preference index i.e., ($T_{novel}/(T_{novel} + T_{familiar}) \times 100$).

## Contextual fear conditioning

The test mice were divided into two groups: the fear group (receiving foot shock) and the control group (not receiving foot shock). The experiment was carried out in 2 days. On the first day (training phase), the mice were placed inside the fear conditioning chamber ($32 \times 28 \times 30$ cm) with a metal grid floor for 3 min. The fear group received two foot shocks of 0.5 mA for 2 s at 120 s and 150 s. The control group was also placed in the box for 3 min but did not receive any foot shock. After 3 min the test mice were returned to their home cage. On the second day, both the control group and the fear group were placed in the fear conditioning chamber for 3 min. At this time no foot shock was delivered to either group. Each session was video-recorded and analyzed for freezing behavior. Freezing time was calculated and represented as the freezing percentage.

## Barnes maze test

The test lasted for 6 days and was divided into 3 phases: adaptation session on day 0, spatial acquisition trial from day 1 to 4 and probe trial on day 5. The test mouse was placed in the center of a gray circular platform having 20 evenly spaced holes on its perimeter. A bright light of 1200 lux was used as a negative reinforcement stimulus that motivated mice to locate the escape box which was placed underneath one of the 20 holes. During the adaptation session, each test mouse was placed on the platform and allowed to explore the arena for 5 min to locate the hole with the escape box. If the mouse failed to locate the escape box it was gently guided to the escape box. Next, for the spatial-acquisition session, the mice were placed on the platform for 5 min and allowed to find the escape box. The escape box was immediately closed after the mouse entered the box and allowed the mice to stay in the box for 2 min. This helps them to associate the escape box as a safe place. Three consecutive trials were done each day for 4 days. The probe trial was done on day 5 where the escape box under the target hole was removed and the test animal was allowed to explore the arena for 2 min under bright light as above. All the trials were video recorded and analyzed using ImageJ software. The latency to find the escape box and the number of errors made while locating the escape box were calculated for all the special-acquisition trials. For assessment of the probe trial, the circular platform was divided into 4 quadrants and time spent by the mice in the target quadrant (quadrant with the initial target hole) was calculated and represented as target quadrant occupancy.

## Experimental design and statistical analysis

Student's $t$-test was used when two conditions or groups were compared. Two-way ANOVA with post hoc Tukey or Šidák's HSD test was used when making multiple comparisons. Two-tailed Mann–Whitney test was performed as non-parametric test to compare two groups when the criteria for Student's t-test was not met. Due to the design of our research, no blinding was performed.

## This paper explained

### Problem

Fragile X syndrome (FXS) is caused by the lack of fragile X messenger ribonucleoprotein (FMRP) that is encoded by the *Fmr1* gene. FXS patients and the mouse model of FXS, the *Fmr1* KO mice, all exhibit excessive protein synthesis, which is central to most disease-specific molecular and behavioral defects in FXS. However, there remains no effective treatment for FXS.

### Results

Here, we showed that a positive allosteric modulator for mGluR7, AMN082, effectively represses protein synthesis by reducing phosphorylation of ERK1/2 and eIF4E. This translational suppressive effect appears to be *Fmr1*-independent. We further showed that treatments of AMN082 reduce neuronal excitability and susceptibility to audiogenic seizures in *Fmr1* KO mice. Lastly, treatments of AMN082 alleviate repetitive behavior and improve learning and memory in *Fmr1* KO mice.

### Impact

Our results uncover the novel roles of mGluR7 and AMN082 in translational control and suggest activation of mGluR7 as a potential therapeutic approach for treating FXS. Given that FXS presents with symptoms that are common to many other neurological and psychiatric disorders, our findings also introduce mGluR7 as a novel therapeutic target for other diseases that are associated with uncontrolled protein synthesis.

Specific sample numbers, including the numbers of cells or repeats, are indicated in the figure legends. The sample size was estimated by G*Power 3.1. No samples or animals were excluded from our analyses. The data presented in this study have been tested for normality using the Kolmogorov–Smirnov test. Data analyses were performed using GraphPad Prism software. Differences are considered significant when $p < 0.05$.

More information about FXS can be found at FRAXA Research Foundation website (https://www.fraxa.org/).

## Data availability

This study includes no data deposited in external repositories.

## Peer review information

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

## Acknowledgements

This work is supported by National Institute of Health R01MH124827 and R21NS130751 to N-PT as well as a FRAXA Research Foundation Postdoctoral Fellowship to VK.

## Author contributions

**Vipendra Kumar**: Conceptualization; Data curation; Formal analysis; Funding acquisition; Validation; Investigation; Methodology; Writing—original draft; Project administration; Writing—review and editing. **Kwan Young Lee**: Data curation; Formal analysis; Investigation; Methodology. **Anirudh Acharya**: Data curation; Formal analysis. **Matthew S Babik**: Data curation. **Catherine A Christian-Hinman**: Resources; Methodology. **Justin S Rhodes**: Resources; Methodology. **Nien-Pei Tsai**: Conceptualization; Supervision; Funding acquisition; Investigation; Writing—original draft; Project administration; Writing—review and editing.

## Disclosure and competing interests statement

The authors declare no competing interests.

# Expanded View Figure

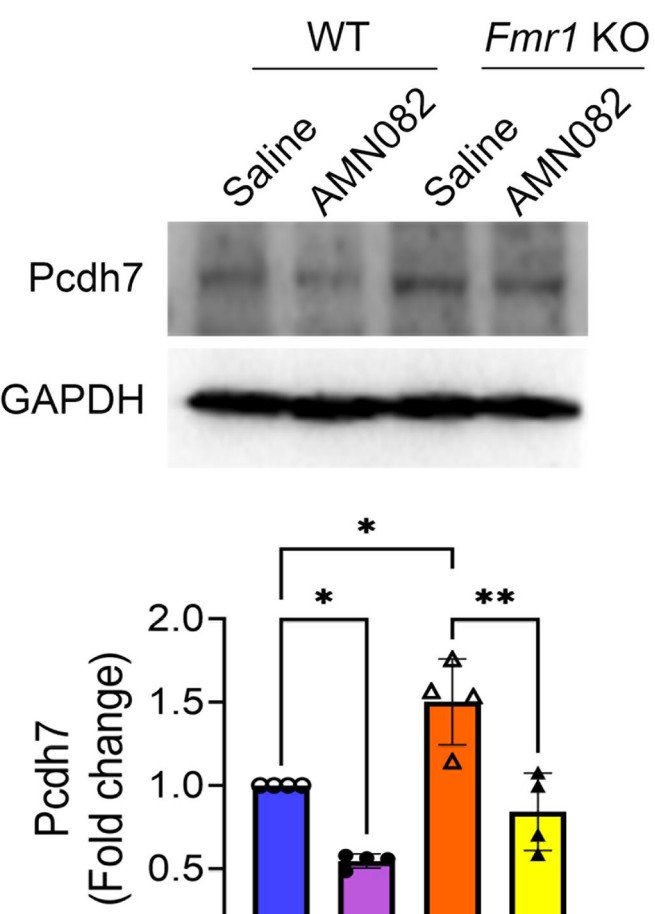

**Figure EV1.   Activation of mGluR reduced the levels of an Fmrp target protein Pcdh7 in the WT and *Fmr1* KO hippocampus.**

Top: Representative blot showing expression of Pcdh7 in the hippocampal lysates of 6–8-week-old WT and *Fmr1* KO mice injected with saline or AMN082 (1 mg/kg). Bottom: Plot showing the quantification of band intensities of Pcdh7 from 4 independent sets of experiments expressed as fold change. Data were analyzed using Two-way ANOVA with Tukey's test and presented as mean +/- SEM. WT +Saline vs WT + AMN082, $p = 0.0213$; WT+Saline vs *Fmr1*KO+Saline, $p = 0.0120$; *Fmr1*KO+Saline vs *Fmr1*KO + AMN082, $p = 0.0021$. Source data are available online for this figure.

