## [Peer Review File · EMBO Molecular Medicine]

mGluR7 allosteric modulator AMN082 corrects protein synthesis and pathological phenotypes in FXS

Vipendra Kumar, Kwan Young Lee, Anirudh Acharya, Matthew Babik, Catherine Christian-Hinman, Justin Rhodes, and Nien-Pei Tsai

Corresponding author: Nien-Pei Tsai (nptsai@illinois.edu)

Review Timeline:

Submission Date:	5th Aug 23
Editorial Decision:	6th Sep 23
Revision Received:	20th Dec 23
Editorial Decision:	31st Jan 24
Revision Received:	31st Jan 24
Accepted:	6th Feb 24

Editor: Poonam Bheda

Transaction Report:

6th Sep 2023

Dear Dr. Tsai,

Thank you for the submission of your manuscript to EMBO Molecular Medicine. We have now received feedback from the three reviewers who agreed to evaluate your manuscript. As you will see from the reports below, the referees acknowledge the interest of the study and are overall supporting publication of your work pending appropriate revisions.

Addressing the reviewers' concerns in full will be necessary for further considering the manuscript in our journal, and acceptance of the manuscript will entail a second round of review. EMBO Molecular Medicine encourages a single round of revision only and therefore, acceptance or rejection of the manuscript will depend on the completeness of your responses included in the next, final version of the manuscript. For this reason, and to save you from any frustrations in the end, I would strongly advise against returning an incomplete revision.

We are expecting your revised manuscript within three months, if you anticipate any delay, please contact us.

We require:

4) A .docx formatted letter INCLUDING the reviewers' reports and your detailed point-by-point responses to their comments. As part of the EMBO Press transparent editorial process, the point-by-point response is part of the Review Process File (RPF), which will be published alongside your paper.

5) A complete author checklist, which you can download from our author guidelines (<https://www.embopress.org/page/journal/17574684/authorguide#submissionofrevisions>). Please insert information in the checklist that is also reflected in the manuscript. The completed author checklist will also be part of the RPF.

6) Please note that all corresponding authors are required to supply an ORCID ID for their name upon submission of a revised manuscript.

7) It is mandatory to include a 'Data Availability' section after the Materials and Methods. Before submitting your revision, primary datasets produced in this study need to be deposited in an appropriate public database, and the accession numbers and database listed under 'Data Availability'. Please remember to provide a reviewer password if the datasets are not yet public (see <https://www.embopress.org/page/journal/17574684/authorguide#dataavailability>).

In case you have no data that requires deposition in a public database, please state so in this section. Note that the Data Availability Section is restricted to new primary data that are part of this study. This study includes no data deposited in external repositories.

8) For data quantification: please specify the name of the statistical test used to generate error bars and P values, the number (n) of independent experiments (specify technical or biological replicates) underlying each data point and the test used to calculate p-values in each figure legend. The figure legends should contain a basic description of n, P and the test applied. Graphs must include a description of the bars and the error bars (s.d., s.e.m.). Please provide exact p values.

9) Our journal encourages inclusion of *data citations in the reference list* to directly cite datasets that were re-used and obtained from public databases. Data citations in the article text are distinct from normal bibliographical citations and should directly link to the database records from which the data can be accessed. In the main text, data citations are formatted as

follows: "Data ref: Smith et al, 2001" or "Data ref: NCBI Sequence Read Archive PRJNA342805, 2017". In the Reference list, data citations must be labeled with "[DATASET]". A data reference must provide the database name, accession number/identifiers and a resolvable link to the landing page from which the data can be accessed at the end of the reference. Further instructions are available at .

13) Author contributions: CRediT has replaced the traditional author contributions section because it offers a systematic machine readable author contributions format that allows for more effective research assessment. Please remove the Authors Contributions from the manuscript and use the free text boxes beneath each contributing author's name in our system to add specific details on the author's contribution. More information is available in our guide to authors.

Please also suggest a striking image or visual abstract to illustrate your article as a PNG file 550 px wide x 300-600 px high. Share synopsis text and image, as well as eTOC:

Please note that these would be the final versions and changes during proofing are usually not allowed

16) As part of the EMBO Publications transparent editorial process initiative (see our Editorial at <http://embomolmed.embopress.org/content/2/9/329>), EMBO Molecular Medicine will publish online a Review Process File (RPF) to accompany accepted manuscripts.

In the event of acceptance, this file will be published in conjunction with your paper and will include the anonymous referee reports, your point-by-point response and all pertinent correspondence relating to the manuscript. Let us know whether you agree with the publication of the RPF and as here, if you want to remove or not any figures from it prior to publication.

I look forward to receiving your revised manuscript.

Yours sincerely,

Poonam Bheda

Poonam Bheda, PhD
Scientific Editor
EMBO Molecular Medicine

**** Reviewer's comments ****

Referee #1 (Comments on Novelty/Model System for Author):

This is discussed in the review

Referee #1 (Remarks for Author):

This is a very interesting and potentially important study to describe a role for a mGluR7 agonist to repress excess protein synthesis resulting in neurophysiological and behavioral phenotypes in a mouse model of Fragile x syndrome. A major strength is the new biological insight and conceptual advance of mGlu7 as potential therapeutic target for Fragile x syndrome and related disorders. The specific strengths of the paper are the convincing physiology and behavior, but there are concerns on the biochemical evidence for effects on downstream signaling and protein synthesis. There are also related concern on the experimental design to analyze the biochemical effects shown in Fig. 1. These concerns are addressable with new analysis and possibly may require new experiments to compare Fmr1 KO and WT side by side with a 2-way ANOVA design to examine for genotype and treatment interactions.

Specific comments -

The Fmr1 KO mouse model has been widely used to demonstrate excess and dysregulated protein synthesis and neuronal signaling. This includes experiments done in cultured neurons, brain slices and synaptoneurosomes. Experiments in Fig. 1D reports that application of mGlu7 agonist, AMN082, decreased overall protein synthesis in both KO and WT cultures, as measured using puromycin incorporation. The KO and WT data are presented separately, analyzed by T-test, and each genotype is normalized to 1.0. This is not the standard way to analyze drug and genotype differences, and importantly fails to allow one to observe the "excess" protein synthesis in FXS and normalizing effects of drug treatment. The data should be replotted, as done in many papers in the field, to analyze four conditions together (WT/vehicle, KO/veh, WT/drug, KO/drug) and use a 2-way ANOVA. This will allow observation of the excess protein synthesis in Fmr1 KO and evaluation of how this is attenuated by mGlu7 agonist. The experiments may need to be repeated if KO and WT were not performed in the same experiment side by side. The goal of any successful drug manipulation is to dampen the moderate excess in FX cells, but not to severely reduce protein synthesis below baseline WT levels.

Similarly in Fig. 1, the authors need to examine the effects of mGlu7 agonist vs. vehicle on elevated basal phosphorylation of ERK and eIF4E, performed side by side, WT vs. KO, not normalized to 1.0, and analyzed by 2-way ANOVA. There are several papers published in the field using this approach.

The experiments showing that eIF4E phosph is inhibited by mGlu7 agonist is interesting but needs to be expanded upon to provide more mechanistic insight. One potential mechanism for drug rescue could be to reduce excess eIF4E and eIF4G interactions (Santini et al. 2017).

Another potential experiment to strengthen the data showing rescue of biochemical phenotypes would be to examine if the expression of FMRP targets is normalized e.g. MAP1b, CaMKIIa, MMP-9, etc. For example, metformin was shown to normalize ERK signaling, eIF4E phosphorylation and expression of MMP-9 in Fmr1 KO mice compared to WT mice littermate controls (see Fig. 2) (Gantois et al. Nat. Med. 2017).

Note that all of the behavioral assessment analyzed WT and KO together using 2-way ANOVA, so this should be carried through for in vitro studies.

Related to the above comments, Fig 4D shows the proposed model for abnormally elevated ERK signaling, phosphorylation of

eIF4E and protein synthesis in FXS, which are attenuated/rescued by application of mGlu7 agonist. Again, the experimental design does not allow one to make this conclusion, because the data in WT and KO are normalized to 1.0, not allowing evaluation of whether or not these phenotypes exist in this model system, and the effects of drug treatment side by side.

Need for in vivo analysis - All of this biochemical assessment is done on primary cortical cultures from postnatal day 0-1 mice. This leaves a disconnect with the behavioral studies to test the mGlu7 agonist done in adult mice. It would be very helpful to have some biochemical analysis of drug efficacy in vivo, as done for targeting p70 S6K (Bhattacharya et al. Neuron 2012; Neuropsychopharm 2016) and targeting PI3K (Gross et al., Cell Reports 2015; Neuropsychopharm 2019).

Other comments -

The standard practice is to use littermate controls, even for culturing postnatal neurons. Please clarify if this method is used.

Fig. 1 implies that mGluR7 is on the postsynaptic side where it is known that there is excess protein synthesis due to elevated and dysregulated ERK. These signaling impairments have not been shown on the presynaptic side. Presynaptic mGluR7 agonist may decrease glutamate release. The model needs to clarify these gaps in understanding pre- vs. post-synaptic MOA.

There was no assessment of cell biological phenotypes, for example, excess dendritic spines in Fmr1 KO. I appreciate these experiments are time consuming and may be beyond the scope of this report. Some comment on whether or not this was examined could be noted for future studies.

A few studies have been published to show reduced cAMP in human FXS cells, which led to animal studies and clinical trials with PDE4 inhibitors to treat FX. Since cAMP is directly downstream of mGlu7, there needs to be some discussion on this topic and whether or not the data fit with that cAMP model.

Referee #2 (Comments on Novelty/Model System for Author):

The study identifies a new strategy that shows therapeutic efficacy in an animal model of Fragile X syndrome.

Referee #2 (Remarks for Author):

The study examined the therapeutic effects of an mGluR7 agonist AMN082 in a mouse model of Fragile X syndrome. Although it is not clear whether the overall mGluR7 signaling (either through 7a or 7b) is altered in FXS, AMN082 showed robust effects on correcting AGS, repetitive behavior, recognition memory, and contextual fear memory in Fmr1 KO mice. AMN082 non-specifically reduced protein synthesis, ERK1/2-eIF4E signaling, and neuronal excitability in both wild-type and FXS neurons. Surprisingly, the study did not detect hyperlocomotion and social interaction deficits in the Fmr1 KO mice. Most of the experiments are well-designed and appropriate. The data reasonably support the conclusion. The results suggest a new therapeutic approach, the molecular target of which is not FXS-specific.

Specific concerns and comments are listed below.

1. Only the drug effects on the level of pERK1/2, p-mTOR, p-eIF4E, and p-eIF2a are reported; genotype effects also need to be disclosed. Two-way ANOVA rather than a t-test would be more appropriate to determine both genotype and drug effect. Similarly, genotype effects on protein synthesis are not clearly disclosed and directly compared between wild-type and KO neurons.
2. While some of the behavior outcomes depend on hippocampus function, mGluR7 expression, protein synthesis, and ERK1/2-eIF4E were mainly examined with cortical neurons.
3. Activation of mGluR7 causes Gi activation, leading to inhibition of adenylyl cyclases and in turn suppresses ERK1/2. This therapeutic approach is in line with others that involve the inhibition of adenylyl cyclase or ERK1/2. However, there are also strategies that involve enhancing cAMP (e.g., through the use of PDE inhibitors). Discussion on these seemingly contradictory strategies is needed.

Referee #3 (Remarks for Author):

The work presented by Kumar and colleagues interrogates the effects of a synthetic organic compound, AMN082, in a mouse model of the Fragile X Syndrome (FXS), the Fmr1 KO mouse.

Using a rather broad panel of approaches ranging from biochemistry, immunohistochemistry, electrophysiology on cultured

neurons and a battery of behavioural tests, the authors conclude that AMN082 modifies protein synthesis pathways and, in parallel, some of the behavioural abnormalities of the mouse model.

The lack of effective curative treatments for FXS warrants new ideas and the choice of an understudied glutamate receptor is interesting and makes, in principle, the article original.

In line with the study, Dasgupta et al (PMID 32310084) indirectly showed that group III metabotropic glutamate receptors modulate the ERK/MAPK pathway with repercussions on long-term plasticity in the rat hippocampus. This article is, in my opinion, worth mentioning.

Much more importantly, the receptor subtype in question, mGlu7, has to date no really specific pharmacology, especially if one aims at its activation. In particular, the compound being the subject of the present study is supposedly a positive allosteric modulator specific for mGlu7. However, a series of papers have reported repeatedly important off-target effects of AMN082. These effects, including in vivo on locomotor activity, were ascribed to its affinity for norepinephrine and dopamine transporters, opioid and adrenergic receptors, sometimes greater than for mGlu7. In addition, a metabolite of AMN082 produced upon in vivo administration has a strong affinity for the serotonin transporter SERT (Sukoff-Rizzo et al PMID 21508084, Ahnaou et al. PMID 27211063, Pałucha-Poniewiera et al PMID 23085340). Notwithstanding the potential of the results obtained upon administration of the compound in Fmr1 deficient animals, it is crucial to keep in mind that some (most?) of the activity might not be due to activation of mGlu7, whereas none of the studies questioning the specificity of AMN082 is mentioned in the present article.

Some observations and/or discrepancies seem to have been skimmed over without real discussion (e.g. contrasting results in social behaviour or locomotor activity compared to other studies on the FXS model).

Some puzzling points remain as well. For e.g. on page 6, line 132, the authors state : "Because mGluR7 KO mainly impacts the expression of mGluR7a isoform (Figure 1E)...", while the original article creating the mGluR7 KO mouse showed that the mGluR7b RNA is not detected by RT-PCR using 7b isoform-specific primers (Sansig et al, PMID 11698585).

Overall the article deserves a serious re-assessment of the literature on AMN082 and, perhaps, a simple rephrasing and refocussing of the title? The end results on the mouse phenotype might maintain their interest, whether or not they are obtained with a "specific mGlu7 positive allosteric modulator" or a broad spectrum pharmacological tool...

Minor points:

- Figure 2B: what is the assumption on the effect on action potential firing, is it a pre- or post-synaptic effect? As the modulation is observed on induced firing, does it mean that mGlu7 is postsynaptic in this preparation? What is the effect on spontaneous firing?

Figure 2C: given the number of animals tested (as well as the impossibility to assume a Gaussian distribution for audiogenic seizure scores?), a non-parametric test such as Mann-Whitney test would be more appropriate (and, if I am not wrong, the values showed on the graph have a $p > 0.05$ to be similar).

We would like to thank the reviewers for their constructive comments on our manuscript. As we outlined below, we have repeated or add more data points in our experiments requested by the reviewers (Figs. 1D, 1F, 1G, 1H, 1I and 3C), provided a substantial amount of new data (Figs 1J, 2A-2D, and EV1) and carefully revised the text suggested by the reviewers. The changes in manuscript are marked in RED. We believe the manuscript is now greatly improved as a consequence of the reviewers' suggestions. In the following, we have provided point-by-point responses to reviewers' comments in BLUE.

Referee #1 (Remarks for Author):

This is a very interesting and potentially important study to describe a role for a mGluR7 agonist to repress excess protein synthesis resulting in neurophysiological and behavioral phenotypes in a mouse model of Fragile x syndrome. A major strength is the new biological insight and conceptual advance of mGlu7 as potential therapeutic target for Fragile x syndrome and related disorders. The specific strengths of the paper are the convincing physiology and behavior, but there are concerns on the biochemical evidence for effects on downstream signaling and protein synthesis. There are also related concern on the experimental design to analyze the biochemical effects shown in Fig. 1. These concerns are addressable with new analysis and possibly may require new experiments to compare *Fmr1* KO and WT side by side with a 2-way ANOVA design to examine for genotype and treatment interactions.

Specific comments -

The *Fmr1* KO mouse model has been widely used to demonstrate excess and dysregulated protein synthesis and neuronal signaling. This includes experiments done in cultured neurons, brain slices and synaptoneurosomes. Experiments in Fig. 1D reports that application of mGlu7 agonist, AMN082, decreased overall protein synthesis in both KO and WT cultures, as measured using puromycin incorporation. The KO and WT data are presented separately, analyzed by T-test, and each genotype is normalized to 1.0. This is not the standard way to analyze drug and genotype differences, and importantly fails to allow one to observe the "excess" protein synthesis in FXS and normalizing effects of drug treatment. The data should be replotted, as done in many papers in the field, to analyze four conditions together (WT/vehicle, KO/veh, WT/drug, KO/drug) and use a 2-way ANOVA. This will allow observation of the excess protein synthesis in *Fmr1* KO and evaluation of how this is attenuated by mGlu7 agonist. The experiments may need to be repeated if KO and WT were not performed in the same experiment side by side. The goal of any successful drug manipulation is to dampen the moderate excess in FX cells, but not to severely reduce protein synthesis below baseline WT levels.

As suggested by the reviewer, we have repeated the experiments in Fig. 1 (Figs. 1D, 1F, 1G, 1H, 1I and new 1J) so that four conditions were performed at the same time and statistically analyzed together by a 2-way ANOVA. New data confirmed our previous conclusion that AMN082 inhibits protein synthesis in both WT and *Fmr1* KO neurons. With four conditions together, we also confirmed that the protein synthesis in *Fmr1* KO neurons treated with AMN082 does not go below baseline WT levels.

Similarly in Fig. 1, the authors need to examine the effects of mGlu7 agonist vs. vehicle on elevated basal phosphorylation of ERK and eIF4E, performed side by side, WT vs. KO, not normalized to 1.0, and analyzed by 2-way ANOVA. There are several papers published in the field using this approach.

These experiments (Fig. 1F and 1H) have also been repeated and the data have been revised to compare all four conditions with a 2-way ANOVA.

The experiments showing that eIF4E phosph is inhibited by mGlu7 agonist is interesting but needs to be expanded upon to provide more mechanistic insight. One potential mechanism for drug rescue could be to reduce excess eIF4E and eIF4G interactions (Santini et al. 2017).

In the revision, we assessed the interaction between eIF4E and eIF4G using m⁷GTP beads, as previously performed (Santini et al, 2017), in both WT and *Fmr1* KO neurons following AMN082 treatment. As shown (new Fig. 1J), AMN082 significantly reduces eIF4E and eIF4G interactions, which is consistent with our data showing that AMN082 treatment leads to eIF4E dephosphorylation (Fig. 1H).

Another potential experiment to strengthen the data showing rescue of biochemical phenotypes would be to examine if the expression of FMRP targets is normalized e.g. MAP1b, CaMKIIa, MMP-9, etc. For example, metformin was shown to normalize ERK signaling, eIF4E phosphorylation and expression of MMP-9 in *Fmr1* KO mice compared to WT mice littermate controls (see Fig. 2) (Gantois et al. Nat. Med. 2017).

To address this suggestion, we first chose MMP9. However, we encountered significant challenges with all three antibodies that we purchased (Sigma #AB19016, Torrey Pines Biolabs #TP221 and Abcam #ab283575). These antibodies display signals even in MMP9 knockout lysates. We then turned to another FMRP target: protocadherin-7 (*Pcdh7*) (Darnell et al, 2011), which we studied before (Liu et al, 2017). As shown (Fig. EV1), we observed a higher level of *Pcdh7* in *Fmr1* KO hippocampi when compared to WT hippocampi. Importantly, treatment of AMN082 reduces the expression of *Pcdh7* in both WT and *Fmr1* KO mice.

Note that all of the behavioral assessment analyzed WT and KO together using 2-way ANOVA, so this should be carried through for in vitro studies.

Related to the above comments, Fig 4D shows the proposed model for abnormally elevated ERK signaling, phosphorylation of eIF4E and protein synthesis in FXS, which are attenuated/rescued by application of mGlu7 agonist. Again, the experimental design does not allow one to make this conclusion, because the data in WT and KO are normalized to 1.0, not allowing evaluation of whether or not these phenotypes exist in this model system, and the effects of drug treatment side by side.

With the newly added or repeated experiments in Fig-1, we believe the model is now much more supported by our findings.

Need for in vivo analysis - All of this biochemical assessment is done on primary cortical cultures from postnatal day 0-1 mice. This leaves a disconnect with the behavioral studies to test the mGlu7 agonist done in adult mice. It would be very helpful to have some biochemical analysis of drug efficacy in vivo, as done for targeting p70 S6K (Bhattacharya et al. Neuron 2012; Neuropsychopharm 2016) and targeting PI3K (Gross et al., Cell Reports 2015; Neuropsychopharm 2019).

As suggested by the reviewer, we have conducted the experiments by intraperitoneally injecting AMN082 or saline in WT or *Fmr1* KO mice. As shown (Figs. 2 A-D), we observed reduced protein synthesis and reduced phosphorylation of ERK1/2 and eIF4E in hippocampal lysates of both WT and *Fmr1* KO mice following the injections of AMN082. These results support our results from cell cultures *in vitro* and strengthened our conclusion.

Other comments -

The standard practice is to use littermate controls, even for culturing postnatal neurons. Please clarify if this method is used.

We did not use littermate controls from heterozygous breeding pairs. Our reasoning of using separate WT and *Fmr1* KO breeding pairs is to minimize the number of mice used in each project and the cost/resources/manpower for constant genotyping. However, in order to minimize the concern of variability between animals, we used littermates to make our WT and *Fmr1* KO breeding pairs. We have added the information about our breeding scheme in page 18.

Fig. 1 implies that mGluR7 is on the postsynaptic side where it is known that there is excess protein synthesis due to elevated and dysregulated ERK. These signaling impairments have not been shown on the presynaptic side. Presynaptic mGluR7 agonist may decrease glutamate release. The model needs to clarify these gaps in understanding pre- vs. post-synaptic MOA.

We have modified our model in Fig-2E (previously Fig-1J) to focus on signaling pathway downstream of mGluR7. In page 16 of the manuscript, we have now described a future direction to assess pre- and post-synaptic effects of mGluR7 in the translational control.

There was no assessment of cell biological phenotypes, for example, excess dendritic spines in *Fmr1* KO. I appreciate these experiments are time consuming and may be beyond the scope of this report. Some comment on whether or not this was examined could be noted for future studies.

We thank the reviewer for this suggestion. We have now added dendritic spine imaging as a potential future direction in page 16.

A few studies have been published to show reduced cAMP in human FXS cells, which led to animal studies and clinical trials with PDE4 inhibitors to treat FX. Since cAMP is directly downstream of mGlu7, there needs to be some discussion on this topic and whether or not the data fit with that cAMP model.

We have added discussion regarding PDE4 and cAMP in page 16. In brief, while PDE4 inhibitors elevate cAMP and mGluR7 potentially reduces cAMP production, our findings are consistent with other preclinical studies in which FXS symptoms are eased when ERK1/2 signaling is dampened (Sethna et al, 2017). Our new discussion describes the existing controversy regarding cAMP levels in FXS.

Referee #2 (Comments on Novelty/Model System for Author):

The study identifies a new strategy that shows therapeutic efficacy in an animal model of Fragile X syndrome.

Referee #2 (Remarks for Author):

The study examined the therapeutic effects of an mGluR7 agonist AMN082 in a mouse model of Fragile X syndrome. Although it is not clear whether the overall mGluR7 signaling (either through 7a or 7b) is altered in FXS, AMN082 showed robust effects on correcting AGS, repetitive behavior, recognition memory, and contextual fear memory in *Fmr1* KO mice. AMN082 non-specifically reduced protein synthesis, ERK1/2-eIF4E signaling, and neuronal excitability in both wild-type and FXS neurons. Surprisingly, the study did not detect hyperlocomotion and social interaction deficits in the *Fmr1* KO mice. Most of the experiments are well-designed and appropriate. The data reasonably support the conclusion. The results suggest a new therapeutic approach, the molecular target of which is not FXS-specific.

Specific concerns and comments are listed below.

1. Only the drug effects on the level of pERK1/2, p-mTOR, p-eIF4E, and p-eIF2a are reported; genotype effects also need to be disclosed. Two-way ANOVA rather than a t-test would be more appropriate to determine both genotype and drug effect. Similarly, genotype effects on protein synthesis are not clearly disclosed and directly compared between wild-type and KO neurons.

This concern was also raised by Reviewer 1. In the revision, we have repeated our experiments in Fig-1 (Figs. 1D, 1F, 1G, 1H, 1I and new 1J) to have all four conditions performed and statistically compared using a 2-way ANOVA.

2. While some of the behavior outcomes depend on hippocampus function, mGluR7 expression, protein synthesis, and ERK1/2-eIF4E were mainly examined with cortical neurons.

We use cortical neuron cultures in order to obtain sufficient amount of materials for biochemical experiments. In the revision, we have performed new experiments (Fig. 2A-D) to assess protein synthesis and signaling pathways downstream of mGluR7 in hippocampus *in vivo*. The new results were consistent with our data in cultured neurons (Fig. 1).

3. Activation of mGluR7 causes Gi activation, leading to inhibition of adenylyl cyclases and in turn suppresses ERK1/2. This therapeutic approach is in line with others that involve the inhibition of adenylyl cyclase or ERK1/2. However, there are also strategies that involve enhancing cAMP (e.g., through the use of PDE inhibitors). Discussion on these seemingly contradictory strategies is needed.

This was also brought up by Reviewer 1. We have included new discussion in page 16 to discuss the controversy regarding cAMP levels.

Referee #3 (Remarks for Author):

The work presented by Kumar and colleagues interrogates the effects of a synthetic organic compound, AMN082, in a mouse model of the Fragile X Syndrome (FXS), the *Fmr1* KO mouse.

Using a rather broad panel of approaches ranging from biochemistry, immunohistochemistry, electrophysiology on cultured neurons and a battery of behavioural tests, the authors conclude that AMN082 modifies protein synthesis pathways and, in parallel, some of the behavioural abnormalities of the mouse model.

The lack of effective curative treatments for FXS warrants new ideas and the choice of an understudied glutamate receptor is interesting and makes, in principle, the article original. In line with the study, Dasgupta et al (PMID 32310084) indirectly showed that group III metabotropic glutamate receptors modulate the ERK/MAPK pathway with repercussions on long-term plasticity in the rat hippocampus. This article is, in my opinion, worth mentioning.

We have now included this paper in our introduction in pages 3-4.

Much more importantly, the receptor subtype in question, mGlu7, has to date no really specific pharmacology, especially if one aims at its activation. In particular, the compound being the subject of the present study is supposedly a positive allosteric modulator specific for mGlu7. However, a series of papers have reported repeatedly important off-target effects of AMN082. These effects, including *in vivo* on locomotor activity, were ascribed to its affinity for norepinephrine and dopamine transporters, opioid and adrenergic receptors, sometimes greater than for mGlu7. In addition, a metabolite of AMN082 produced upon *in vivo* administration has a strong affinity for the serotonin transporter SERT (Sukoff-Rizzo et al PMID 21508084, Ahnaou et al. PMID 27211063, Palucha-Poniewiera et al PMID 23085340). Notwithstanding the potential of the results obtained upon administration of the compound in *Fmr1* deficient animals,

it is crucial to keep in mind that some (most?) of the activity might not be due to activation of mGlu7, whereas none of the studies questioning the specificity of AMN082 is mentioned in the present article.

To support the conclusion that our observation is most likely through mGluR7, our data in Fig. 1E showed that AMN082 did not elicit effects on protein synthesis in cultures made from mGluR7 KO mice. In the revision, we have added another set of data (Fig. 2B) and confirmed no significant effect on protein synthesis in hippocampus following injections of AMN082 in mGluR7 KO mice. But we do agree with the reviewer that the specificity of AMN082 remains a concern. In the revision, we have included new discussion to discuss about the specificity issues previously observed in AMN082 in page 16 and indicate the need for a more specific agonist of mGluR7 in the future.

Some observations and/or discrepancies seem to have been skimmed over without real discussion (e.g. contrasting results in social behaviour or locomotor activity compared to other studies on the FXS model).

Studies have reported inconsistent results regarding social interaction and open field tests in *Fmr1* KO mice (Saré & Levine, 2016; Smith et al, 2009), which may be caused by the different genetic background of the mice and the different environmental setting in each study. We have acknowledged this discrepancy in page 11.

Some puzzling points remain as well. For e.g. on page 6, line 132, the authors state : "Because mGluR7 KO mainly impacts the expression of mGluR7a isoform (Figure 1E)...", while the original article creating the mGluR7 KO mouse showed that the mGluR7b RNA is not detected by RT-PCR using 7b isoform-specific primers (Sansig *et al*, 2001)(Sansig et al, PMID 11698585).

Our understanding is that the mGluR7 KO mice we obtained are not the same mice generated by Sansig et al. Our mGluR7 KO mice were from Mutant Mouse Resource and Research Center (MMRRC, strain name B6.129P2-*Grm7*^{tm1Dgen}/Mmnc, catalog# MMRRC:011626-UNC), which obtained these mice from Deltagen, Inc. According to the supplier website, mGluR7a isoform was the target of this knockout strain. We have included the strain information about these mice in page 18.

Overall the article deserves a serious re-assessment of the literature on AMN082 and, perhaps, a simple rephrasing and refocussing of the title? The end results on the mouse phenotype might maintain their interest, whether or not they are obtained with a "specific mGlu7 positive allosteric modulator" or a broad spectrum pharmacological tool...

As we indicated above, we believe the effects of AMN082 are, at least for the most parts, through mGluR7 based on our data using mGluR KO mice. But given the concern about its specificity, in addition to the new discussion in page 16, we have changed "activation of mGluR7" to "mGluR7 allosteric modulator AMN082" in the title and similarly in the abstract.

We hope by this way, the emphasis can be more on the drug (AMN082) than the receptor (mGluR7).

Minor points:

- Figure 2B: what is the assumption on the effect on action potential firing, is it a pre- or post-synaptic effect? As the modulation is observed on induced firing, does it mean that mGlu7 is postsynaptic in this preparation? What is the effect on spontaneous firing?

The result on action potential firing can only indicate an effect on neuronal intrinsic excitability, which can be a consequence of reduced protein synthesis in cells.

Because the impaired ERK1/2 signaling and protein synthesis were reported mostly at post-synapses of *Fmr1* KO neurons, we believe the effect of AMN082 is on post-synapses. However, we do not have sufficient information to indicate whether mGluR7 is being activated in pre- or post-synapses. As indicated by Reviewer 1, there is a possibility that mGluR7 modulates glutamate release at pre-synapses, leading to changes of signaling at post-synapses. We have added a new future direction about this in page 16.

We employed MEA to assess spontaneous firing and our data in Fig. 3A suggest that AMN082 also reduces the spontaneous firing.

Figure 2C: given the number of animals tested (as well as the impossibility to assume a Gaussian distribution for audiogenic seizure scores?), a non-parametric test such as Mann-Whitney test would be more appropriate (and, if I am not wrong, the values showed on the graph have a $p > 0.05$ to be similar).

We have included more animals in the experiment and employed the Mann-Whitney test to re-assess our data in Fig. 3C (previous Fig. 2C). The data have been updated and the results confirmed an effect of AMN082 on reducing seizure scores in *Fmr1* KO mice.

References:

Darnell JC, Van Driesche SJ, Zhang C, Hung KY, Mele A, Fraser CE, Stone EF, Chen C, Fak JJ, Chi SW, Licatalosi DD, Richter JD, Darnell RB (2011) FMRP stalls ribosomal translocation on mRNAs linked to synaptic function and autism. *Cell* **146**: 247-261

Liu DC, Seimetz J, Lee KY, Kalsotra A, Chung HJ, Lu H, Tsai NP (2017) Mdm2 mediates FMRP- and Gp1 mGluR-dependent protein translation and neural network activity. *Human molecular genetics* **26**: 3895-3908

Santini E, Huynh TN, Longo F, Koo SY, Mojica E, D'Andrea L, Bagni C, Klann E (2017) Reducing eIF4E-eIF4G interactions restores the balance between protein synthesis and actin dynamics in fragile X syndrome model mice. *Science signaling* **10**

Saré RM, Levine M (2016) Behavioral Phenotype of *Fmr1* Knock-Out Mice during Active Phase in an Altered Light/Dark Cycle. **3**

Sethna F, Feng W, Ding Q, Robison AJ, Feng Y, Wang H (2017) Enhanced expression of ADCY1 underlies aberrant neuronal signalling and behaviour in a syndromic autism model. *Nature communications* **8**: 14359

Smith CB, Eadie BD, Zhang WN, Boehme F, Gil-Mohapel J, Kainer L, Simpson JM, Christie BR (2009) *Fmr1* knockout mice show reduced anxiety and alterations in neurogenesis that are specific to the ventral dentate gyrus. *eNeuro* **36**: 361-373

31st Jan 2024

Dear Dr. Tsai,

Thank you for the submission of your revised manuscript to EMBO Molecular Medicine. We have now received the enclosed reports from the referees that were asked to re-assess it. As you will see the reviewers are now globally supportive and I am pleased to inform you that we will be able to accept your manuscript pending the following final amendments:

- 1) Regarding the manuscript type, please resubmit the paper as an Article and not a Report. This is based on the length and number of findings, and Reports are limited to 3 figures.
- 2) In the main manuscript file, please do the following:
 - Reduce keywords to max. 5.
 - Please rename "Conflict of Interest" to "Disclosure and competing interests statement". We updated our journal's competing interests policy in January 2022 and request authors to consider both actual and perceived competing interests. Please review the policy <https://www.embopress.org/competing-interests> and update your competing interests if necessary.
- 3) In the Materials and Methods, please take care of the following:
 - Ethics statement: The ethics statement appears to be duplicated as it is present in both its own section and as well at the beginning of the Animals section in the Materials & Methods. It is not necessary to have a separate section on this, but it is your discretion in which section you keep the ethics statement as long as it appears in the Materials & Methods.
 - Animals: Please also ensure that the gender of the animals involved in experiments is reported.
 - Primary cultures: Given the cells were cultured for 14-16 divisions, please include a sentence in the Materials and Methods as to whether or not the cell lines were tested for mycoplasma contamination. Please also update the Author checklist for this section to state that the information is available in the Materials & Methods.
 - Antibodies: please ensure that company name, catalog number, and dilutions/amounts of each antibody are reported. Currently this information is incomplete in the Materials & Methods sections on western blotting and immunohistochemistry
- 4) Please place individual sections of the manuscript in the following order: Title page - Abstract & Keywords - Introduction - Results - Discussion - Materials & Methods - Data Availability - Acknowledgements - Disclosure and Competing Interests Statement - The Paper Explained - For More Information - References - Figure Legends - Expanded View Figure Legends.
 - Please move the Data Availability section to the end of the Materials and Methods section.
 - Please add "The Paper Explained" to the main manuscript text, rather than submitting as a separate file.
 - For more information: This space should be used to list relevant web links for further consultation by our readers. Could you identify some relevant ones and provide such information as well? Some examples are patient associations, relevant databases, OMIM/proteins/genes links, author's websites, etc...
- 5) For the figures and figure legends, please take care of the following:
 - Please note that a separate 'Data Information' section is required in the legends of all figures. This is currently missing for the legends of figures 1a-j; 2a-d; 3a-c; 4a-c; 5a-c.
 - Please note that we require exact p-values to be reported. Currently exact p-values are not provided.
- 6) Please check your synopsis text and image before submission with your revised manuscript. Please be aware that in the proof stage minor corrections only are allowed (e.g., typos).
- 7) As part of the EMBO Publications transparent editorial process initiative (see our policy here: https://www.embopress.org/transparent-process#Review_Process), EMBO Molecular Medicine will publish online a Peer Review File (PRF) to accompany accepted manuscripts. This file will be published in conjunction with your paper and will include the anonymous referee reports, your point-by-point response and all pertinent correspondence relating to the manuscript. Let us know whether you agree with the publication of the PRF and as here, if you want to remove or not any figures from it prior to publication. Please note that the Authors checklist will be published at the end of the PRF.
- 8) Please provide a point-by-point letter INCLUDING my comments as well as the reviewer's reports and your detailed responses (as Word file).

I look forward to reading a new revised version of your manuscript as soon as possible.

Yours sincerely,

Poonam Bheda

Poonam Bheda, PhD
Scientific Editor
EMBO Molecular Medicine

*** Instructions to submit your revised manuscript ***

***** Reviewer's comments *****

Referee #1 (Comments on Novelty/Model System for Author):

no concerns

Referee #1 (Remarks for Author):

The authors have been very responsive to perform several new experiments and alternative data analysis as suggested to address previous concerns. They also have made many changes to the text and discussion as suggested. This revised manuscript is now excellent and suitable for publication.

Referee #2 (Comments on Novelty/Model System for Author):

The main conclusion is supported by rigorous data that involve tests with molecular, cellular, pharmacological, and behavioral approaches. The identified molecular target and pharmacological intervention are novel. As there is no effective medical treatment for Fragile X syndrome, inhibition of mGluR7 as a potential therapy has a significant impact. The mouse model (i.e., Fmr1 KO mice) is robust and recapitulates many key pathological aspects of Fragile X syndrome.

Referee #2 (Remarks for Author):

With the new addition of data and discussion, the authors have adequately addressed the concerns/comments. I recommend publication.

Referee #3 (Remarks for Author):

The changes and additions provided by the authors make the manuscript, in my opinion, suitable for publication.

The authors addressed the minor editorial issues.

6th Feb 2024

Dear Dr. Tsai,

We are pleased to inform you that your manuscript is accepted for publication and is now being sent to our publisher to be included in the next available issue of EMBO Molecular Medicine.

Yours sincerely,

Poonam Bheda, PhD
Scientific Editor
EMBO Molecular Medicine
